

# 1 Recent Precipitation Decrease Across the Western Greenland Ice Sheet
# 2 Percolation Zone

Lewis, Gabriel[1]; Osterberg, Erich[1]; Hawley, Robert[1]; Marshall, Hans Peter[2]; Meehan, Tate[2]; Graeter, Karina[3];
McCarthy, Forrest[4]; Overly, Thomas[5,6]; Thundercloud, Zayta[1]; Ferris, David[1]
[1]Department of Earth Sciences, Dartmouth College, Hanover, NH, USA
[2]Geosciences Department, Boise State University, Boise, ID, USA
[3]Office of Sustainability, University of Maine, Orono, ME, USA
[4]College of Fisheries and Ocean Sciences, University of Alaska Fairbanks, Fairbanks, AK, USA
[5]NASA Cryospheric Sciences Laboratory, NASA Goddard Space Flight Center, Greenbelt, MD, USA
[6]Earth System Science Interdisciplinary Center (ESSIC), University of Maryland, College Park, MD, USA
*Correspondence to*: Gabriel Lewis (Gabriel.M.Lewis.GR@dartmouth.edu)
**Abstract**
The mass balance of the Greenland Ice Sheet (GrIS) in a warming climate is of critical interest to scientists
and the general public in the context of future sea-level rise. Increased melting in the GrIS percolation zone
due to atmospheric warming over the past several decades has led to increased mass loss at lower elevations.
Previous studies have hypothesized that this warming is accompanied by a precipitation increase, as would
be expected from the Clausius-Clapeyron relationship, negating some of the melt-induced mass loss
throughout the Western GrIS. This study tests that hypothesis by calculating snow accumulation rates and
trends across the Western GrIS percolation zone, providing new critical accumulation estimates in regions
with sparse and/or dated *in situ* data for calibration of future regional climate models. We present
accumulation records from sixteen 22 – 32 m long firn cores and 4436 km of ground penetrating-radar,
covering the past 20 – 60 years of accumulation, collected across the Western GrIS percolation zone as part
of the Greenland Traverse for Accumulation and Climate Studies (GreenTrACS) project. Trends from both
radar and firn cores, as well as commonly used regional climate models, show decreasing accumulation and
precipitation over the 1996 – 2016 period, which we attribute to shifting storm-tracks related to stronger
atmospheric summer blocking over Greenland. Changes in atmospheric circulation over the past 20 years,
specifically anomalously high summertime blocking, have reduced GrIS surface mass balance through both
an increase in surface melting and a decrease in accumulation.

## 29 1. Introduction

Greenland Ice Sheet (GrIS) mass loss has accelerated over the past few decades, with modern mass loss rates
more than double that from Antarctica (van den Broeke et al., 2016). The 2010-2018 GrIS mass loss is 286
$\pm$ 20 Gt a$^{-1}$ (Mouginot et al., 2019), contributing 0.7 $\pm$ 0.2 mm a$^{-1}$ of sea level rise. Over the past 20 years, the
largest warming rates (Hanna et al., 2012) and fastest mass loss have occurred in Western Greenland (26 $\pm$ 7



GT a$^{-2}$ in basins F + G of Sasgen et al., 2012), where surface mass balance (SMB) has decreased between
31.1% (European Centre for Medium Range Weather Forecasting downscaled; ECMWFd) and 76.5%
(Modèle Atmosphérique Régional; MAR) over the 1996 – 2008 period (Vernon et al., 2013) due to higher
surface melt and runoff (van den Broeke et al., 2009, 2016). Modern surface melt rates are at their highest
levels of at least the last 450 years across Western Greenland (Graeter et al., 2018) and more broadly
throughout Greenland (Trusel et al., 2018). In particular, ice core records from Western Greenland show an
abrupt increase in surface melt rates beginning in the middle-late 1990's due to a combination of higher North
Atlantic sea surface temperatures, enhanced summertime blocking highs, and anthropogenic warming
(Graeter et al., 2018).

Enhanced GrIS surface melt is driven fundamentally by rising Greenland temperatures as recorded by
automated weather stations from the Greenland Climate Network (GC-Net; Steffen and Box, 2001), coastal
weather stations (Box, 2002), borehole thermometry (Polashenski et al., 2014), remote sensing (Hall et al.,
2008), and ice core stable isotopes (Buchardt et al., 2012). Average annual temperature across interior
Greenland increased by 0.055 ± 0.044 °C a$^{-1}$ from 2000 – 2012 (Hall et al., 2013), with summer trends
upwards of 0.135 ± 0.047 °C a$^{-1}$ (Hall et al., 2013; Reeves Eyre and Zeng, 2017). These warming trends
extend to the highest elevations of the ice sheet, with warming at Summit Station of 0.09 ± 0.01 °C a$^{-1}$ from
1982 – 2011 (McGrath et al., 2013). Nearly every Greenland dataset shows statistically significant positive
temperature trends in recent decades, especially during the summers (Reeves Eyre and Zeng, 2017).

Basic physics implies that rising temperatures should cause an increase in accumulation over the ice sheet
due to the Clausius-Clapeyron relationship – warmer air has a higher saturation vapor pressure, potentially
leading to more precipitation (Box et al., 2006; Buchardt et al., 2012). The Coupled Model Intercomparison
Project, phase 5 (CMIP5) predicts precipitation increases of 20 – 50% over the GrIS by the end of the 21$^{st}$
century (Bintanja and Selten, 2014), partially offsetting mass loss and sea-level rise from enhanced summer
melt and runoff. However, most *in situ* records of Greenland snow accumulation do not span the modern
period of rapid warming and accelerating mass loss since the mid-1990s. It is difficult to determine whether
accumulation has been increasing with the observed warming temperatures as predicted. For example, the
Program for Arctic Regional Climate Assessment (PARCA) campaign collected 46 shallow ice and firn cores
(<100 m long) and three deeper cores (120 – 152 m long) to capture the spatial and temporal accumulation
variability over the ice sheet (Mosley-Thompson et al., 2001). However, the PARCA firn cores were collected
in 1997 – 1998, just at the onset of accelerated surface melting (Graeter et al., 2018), and the precipitation
record from automated weather stations is too brief and localized to analyze significant precipitation trends
(Rennermalm et al., 2013). The most recently analyzed deep ice cores (over 100 m long) were collected in





2003 – 2004 (D4, D5, Sandy, Katie; Banta and McConnell, 2007) and there have been no published *in situ*
accumulation records from the Western GrIS percolation zone for the past decade. Updated *in situ* snow
accumulation data are needed from this region to assess recent changes in accumulation during this period of
warming and SMB loss from melt and runoff.

In addition to measuring snow accumulation with ice cores and automated snow depth sensors, several
previous studies have used ground-based and airborne radar to calculate GrIS accumulation and trends (e.g.
Medley et al., 2013; Spikes et al., 2004). For example, Hawley et al. (2014) found a 10% increase in northwest
coastal Greenland accumulation over the past 52 years using Ground Penetrating Radar (GPR) along the
Greenland Inland Traverse (GrIT; see Figure 1 for location), although they did not find any statistically
significant trends further inland between the North Greenland Eemian Ice Drilling (NEEM) site and Summit
Station. Similarly, Wong et al. (2015) show a coastal increase in precipitation near Thule over 1981 – 2012,
but no statistically significant change in precipitation rate further inland at the Camp Century, B26, or 2Barrel
ice core sites. Overly et al. (2016) found a 20% accumulation increase below 3000 m a.s.l. on the historic
Expéditions Glaciologiques Internationales au Groenland (EGIG) line from 1994 – 2004 vs. 1985 – 1994
using the Airborne SAR/Interferometric Radar Altimeter System (ASIRAS) radar. We build upon these
previous studies by collecting GPR data across the lower percolation zone of Western Greenland, where
airborne radargrams are often obscured by refrozen melt percolation (Nghiem et al., 2005). By having our
GPR antenna coupled with the snow, we avoid losing energy, and, therefore, penetration depth, from a strong
reflection off of the snow-air interface.

In addition to temperature-precipitation relationships through the Clausius-Clapeyron relationship, previous
studies have analyzed the dynamic climate controls on Greenland precipitation. Mernild et al. (2014), Auger
et al. (2017), and Lewis et al. (2017) have hypothesized that a positive Atlantic Multidecadal Oscillation
(AMO) index correlates with rising accumulation over most of the GrIS interior, since higher sea surface
temperatures increase moisture flux over the GrIS and induce greater snowfall. High pressure (blocking)
systems east of Greenland tend to deflect eastward-moving storms over central Greenland and increase
precipitation, whereas blocking directly over Greenland or in Baffin Bay has the potential to prevent storms
from crossing the ice sheet (Auger et al., 2017). Through reanalysis data, Auger et al. (2017) showed that
persistent blocking highs increase precipitation in southwest Greenland and reduce precipitation in the
southeast.

The Greenland Blocking Index (GBI) quantifies blocking directly over Greenland and is defined as the mean
500 hPa geopotential height for the 60 – 80°N, 20 – 80°W region (Hanna et al., 2016). Over the 1991 – 2015



period there has been an especially high Greenland Blocking Index sustained throughout the summers (Hanna
et al., 2016). Alternatively, persistent blocking episodes have the potential to reduce snowfall accumulation
over the GrIS by displacing the polar jet stream and corresponding storm tracks equatorward, although this
relationship has not yet been documented *in situ*.

Here we develop new accumulation records across the Western GrIS percolation zone using sixteen firn cores
and 4436 km of GPR data collected during an over-ice traverse spanning two field seasons. We evaluate the
veracity of the accumulation data through comparisons of our firn core time series with previous
measurements. We quantify multi-year trends in accumulation across Western Greenland to test the
hypothesis that precipitation has recently increased from the Clausius-Clapeyron relationship and higher GrIS
temperatures. Further, we assess the ability of RCMs to capture the year-to-year variability and multi-year
trends in Western GrIS accumulation. Finally, we evaluate relationships between recent accumulation trends
and atmospheric circulation patterns, particularly changes in storm tracks.

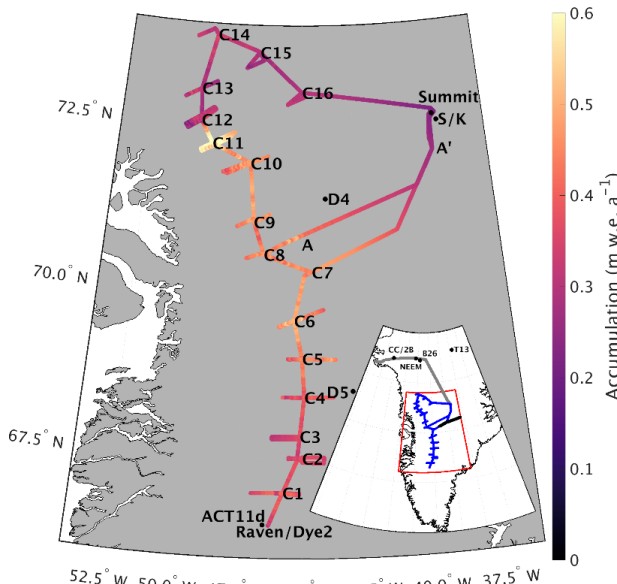


**Figure 1. Average accumulation across the GreenTrACS traverse for the length of each record showing the location of each firn core,**
**ACT11d, D4, D5, Katie (K), Raven/Dye-2, and Sandy (S) ice cores, and Summit Station. Transect A-A' discussed in Section 3.3. Inset**
**shows locations of Camp Century (CC), 2Barrel (2B), NEEM, B26, and TUNU2013 (T13) ice cores, as well as locations of EGIG (black),**
**GrIT (grey), and GreenTrACS (blue) traverses.**
**2. Methods**
This study uses data from the 2016 – 2017 Greenland Traverse for Accumulation and Climate Studies
(GreenTrACS), which measured accumulation and melt across the Western GrIS percolation zone over two



summer snowmobile traverses (closely following the 2150 m a.s.l. elevation contour). The May-June 2016
season traversed 860 km from Raven/Dye-2 northward to Summit Station and the May – June 2017 traverse
made a 1230 km clockwise loop starting and ending at Summit Station (Figure 1). This manuscript focuses
on accumulation rates derived from 400 MHz GPR data collected along the entire traverse path, as well as
sixteen shallow (22 – 32 m long) firn cores spaced 40 – 100 km apart along the backbone of the traverse
(Figure 1). Firn Cores 1 – 7 were collected in 2016 and Cores 8 – 16 were collected in 2017. We returned to
the Core 7 location at the beginning of the 2017 traverse to recover a weather station and to connect the two
season's GPR data. Additionally, we collected GPR data ~30 – 70 km east and west of each core site,
hereafter called "spurs", to measure accumulation changes along strong elevation gradients (see Figure 1).

### 2.1. GPS Positioning

During the 2016 traverse we collected GPS data using a Trimble NetR8 reference receiver with a Zephyr
Geodetic antenna mounted to a Nansen sled ~5 m in front of the GPR antenna. For each spur and the tail
ends of each transect between core sites we performed differential corrections to the GPS data using RTKLIB
2.4.1 and a Trimble NetR8 base station near the core site. Between spurs, when not operating a base station,
we post-processed GPS data in precise point positioning mode (Zumberge et al., 1997). Estimated root-mean-
square horizontal errors were generally between 13 and 18 cm from standard deviations calculated during
stationary periods at the end of spurs. To co-register GPR and GPS data, we used time stamps embedded in
the two data streams and locations where we stopped to save GPR files, approximately every 15 km. The
time drift in the GPR logger is negligible over these durations.

During the 2017 traverse we used GPS data from a Garmin 19x GPS receiver wired directly to the GPR
instrument, which recorded position data at every radar sample with RMS values of 3 m. During radar
processing we average 75 adjacent traces, corresponding to a distance of ~20 m, so errors in GPS positioning
have a negligible effect on the final dataset.

### 2.2. Ground-penetrating radar

We develop a spatially continuous record of accumulation using GPR profiles collected with Geophysical
Survey Systems Inc. (GSSI) SIR-3000 (during 2016) and SIR-30 (during 2017) radar units with a 400 MHz
antenna (following Hawley et al., 2014). The antenna was towed on the snow surface in a small plastic sled
~5 m behind a wooden Nansen sled and ~15 m behind a snow machine. We recorded 2048 samples (2016)
and 4096 samples (2017) per trace over a range window of 800 ns (Figure 2). At a relative permittivity of
$1.26 \pm 0.07$, typical of firn in the GrIS percolation zone, the range was ~111 – 114 m. The 400 MHz short-



pulse radar has a range resolution (ability to resolve distinct features) of 0.35 ± 0.1 m in firn, which is fine
enough to resolve Internal Reflecting Horizons (IRHs) that have been shown to represent isochrones (Medley
et al., 2013; Rodriguez-Morales et al., 2014; Spikes et al., 2004; Hawley et al., 2014). We recorded 10 traces
per second, which at the snowmobile's average travel speed of approximately 2.75 m s$^{-1}$ results in ~3.6 traces
recorded per meter. Note that this spacing between traces varies with vehicle speed.

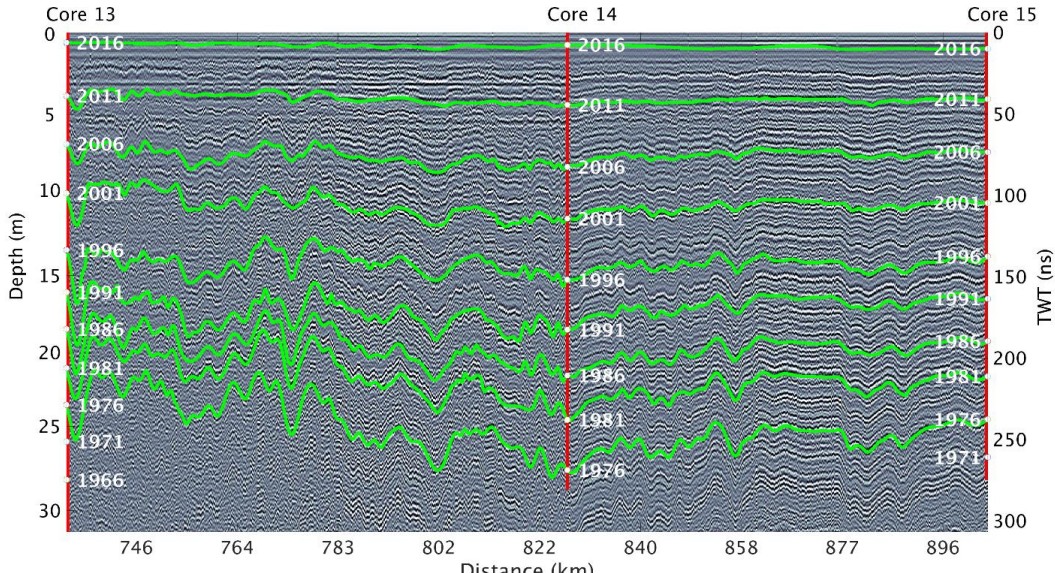


**Figure 2. Radargram showing the top 32 m of the transect along the main 2017 traverse from Core 13 to Core 15. Cores are indicated as red lines down to their final depth, with dates plotted every 5 years at corresponding depths. Traced internal reflecting horizons are shown as isochronous green lines. The depth scale on the vertical axis is calculated from the TWT-depth conversion (see Section 2.4) for Core 13, although there is no visual difference in depth scale across this radargram.**

Depending on signal attenuation within the firn column, IRHs can be traced to a depth of 20 – 50 m (Figure
2), providing accumulation records over the past 20 – 60 years (Figure 3). For areas with high attenuation
(i.e. shallow penetration of the radar signal), such as lower elevation regions with more refrozen melt layers,
we calculate accumulation results for shorter time periods. We are not able to trace as many IRHs to the west
of Cores 10 – 13 compared to the east due to higher signal attenuation, resulting in slightly different average
accumulation values on either side of these cores (Figure 3). Likewise, we experienced an equipment
malfunction at the end of the 2016 traverse, reducing the number of observable IRHs from Core 7 to Summit
Station (Figure 3). We have less confidence in calculated accumulation throughout this section of the traverse
due to this malfunction, although the 2017 Summit to Core 8 interval overlaps nicely with the last 140 km of
the problematic 2016 interval, and provides high quality accumulation measurements for this section near
Summit Station.

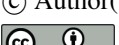


We reduce the GPR data volume and signal noise by averaging 75 adjacent traces, which has the effect of
suppressing random noise by the principle of trace stacking (Yilmaz, 2001). We apply a combination of
median trace filtering, residual mean filtering (Gerlitz et al., 1993), and bandpass filtering using a butterworth
design (Selesnick and Sidney Burrus, 1998) between 200 – 800 MHz. For data visualization, we apply an
automatic gain control (Yilmaz, 2001) to give the interpreter more confidence when picking IRHs.


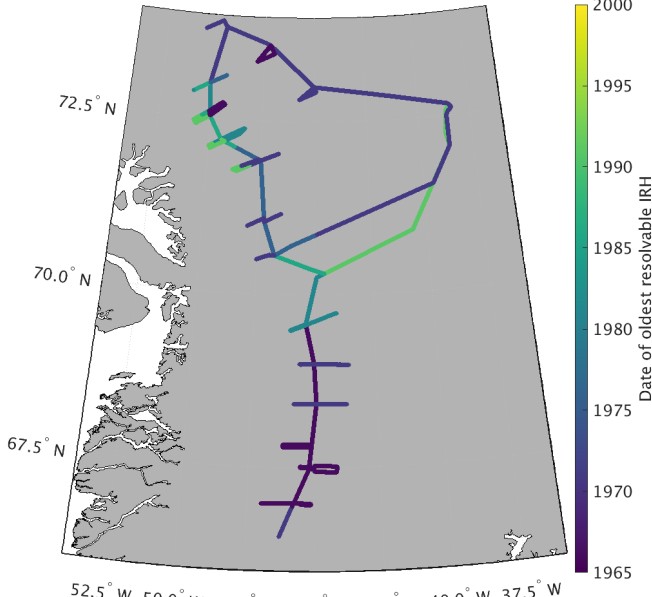


**Figure 3: Date of oldest resolvable internal reflecting horizon throughout the entire GreenTrACS traverse route. Anomalously young**
**ages from Core 7 to Summit are due to equipment malfunction.**

**2.3. Firn core processing and density profiles**
The amount of snow mass and the time span between IRHs are necessary to calculate accumulation rates.
The accumulation rate is a function of the depth-age scale, travel time-depth conversion rate, and the firn
density profile. We obtain the depth-age and depth-density scales from each of the shallow firn cores
collected along the GreenTrACS traverse, and from density models based on temperature and accumulation
rate data.

The sixteen firn cores were drilled using an Ice Drilling Program hand auger with a Kyne sidewinder
attachment (see Graeter et al., 2018). We sampled the firn cores for chemical measurements using a





continuous ice core melter system with discrete sampling (Osterberg et al., 2006). We used an Abakus (Klotz)
laser particle detector to measure microparticle concentrations and size distribution from the continuous ice
core meltwater stream, a Dionex Model ICS5000 capillary ion chromatograph to measure major ion ($Na^+$,
$Mg^{2+}$, $Ca^{2+}$, $K^+$, $NH_4^+$, $Cl^-$, $NO_3^-$, $SO_4^{2-}$) and methanesulfonic acid concentrations, and a Picarro L1102-I and
a Los Gatos Research Liquid Water Isotope Analyzer to measure oxygen and hydrogen isotope ratios ($\delta^{18}O$,
$\delta D$; Graeter et al., 2018).

We determine depth-age curves for each core by identifying annual layers based on seasonal oscillations in
$\delta^{18}O$ and the concentrations of major ions and dust, consistent with previous ice core studies in this region
(Graeter et al., 2018; Mosley-Thompson et al., 2001; Osterberg et al., 2015). We combine the depth-age
scales with measured density to calculate annual accumulation rates at the firn core sites.

At each firn core and at the ends of each spur, we measured the density in the top meter of snow using a 1000
$cm^3$ SnowMetrics cutter. To calculate firn core density profiles, we measured the mass, length, and diameter
of 0.03–1 m long core segments in the field and again after transporting the cores to the Dartmouth College
Ice Core Laboratory. To calculate accumulation rates at Raven/Dye-2, we use density data from a 119.6 m
long firn core collected in 1997 (Bales et al., 2009) and a 19.3 m long core collected from the same location
in 2015, which did not include accumulation data (Vandecrux et al., 2018). For this location we use the most
recent density data for the near-surface and the older densities for depths below the 2015 core. Likewise, we
use a density profile from a 109 m long firn core collected from Summit in 2010 (Mary Albert, personal
communication, 2015). We also incorporate density data from measurements along the EGIG traverse at T19,
T21, T23, T27, and T31 to improve the density profile between Core 7 and Summit (Morris and Wingham,

242   2014).


After collecting each firn core, we measured borehole temperature for 24 – 48 hours using a 20 m long
thermistor string. We estimate mean annual temperature from the deepest thermistor on the twenty-
thermistor-string. For the location of each firn core, we use the depth-density data from that core and then
calculate a Herron and Langway (1980) depth-density model for depths below the core using our measured
mean annual temperature, firn core mean annual accumulation, and top-meter snow density. Likewise, we
calculate Herron-Langway profiles for the ends of each spur using MODIS satellite derived mean annual
temperature (Hall et al., 2012), MAR modeled accumulation (Burgess et al., 2010), and the measured snow
density in the upper meter of each of the spur's snow pits. Finally, we interpolate depth-density profiles both
between firn cores and along radar spurs to estimate the depth-density matrix everywhere along our traverse





(Figure 4). Final accumulation rates are insensitive to the accumulation we use to calculate our Herron-
Langway models (Lewis et al., 2017).

As shown in Figure 4, ice layers within several firn cores are extrapolated laterally along the traverse,
although these dense lenses are typically both localized and heterogeneous in nature (Brown et al., 2011;
Rennermalm et al., 2013). This ice lens density interpolation is as accurate as possible between firn cores
without additional *in situ* data, and this estimation does not significantly alter our results, as discussed in
Section 2.6, since the ice layers represent a small fraction of the total depth to IRHs.

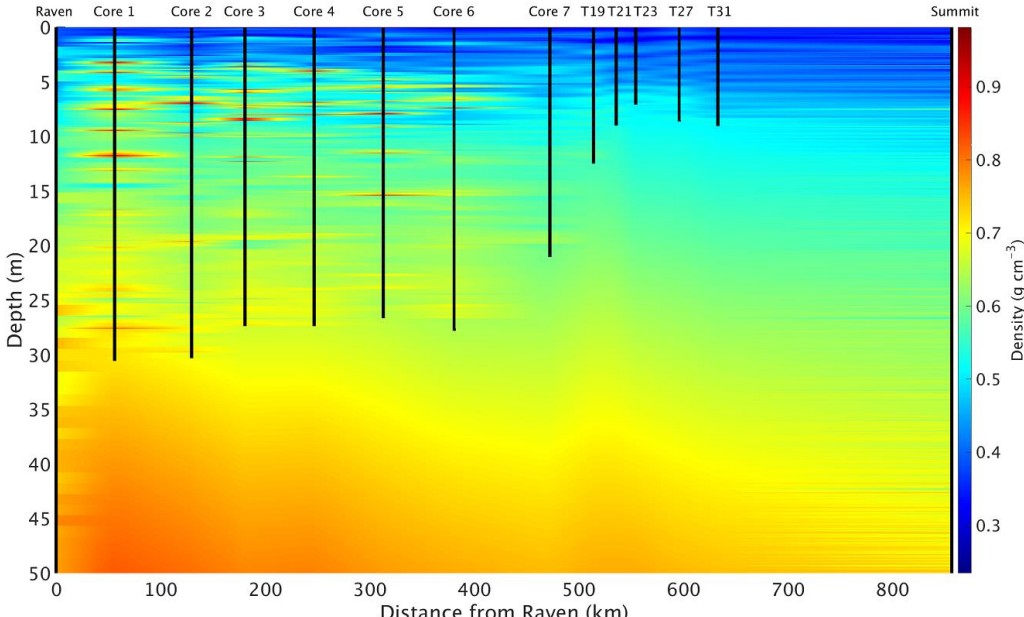


**Figure 4. Depth-density profile along the main 2016 traverse used for calculation of electromagnetic wave velocity and accumulation in**
**this study. Densities are linearly interpolated between the two nearest cores and are modeled using Herron-Langway profiles below the**
**depth of each core. The left and right boundary data come from the Raven/Dye-2 and Summit firn cores, respectively. Ice layers in Cores**
**1 – 5 are clearly visible as red lenses, but their extent is, in reality, likely more localized.**

**2.4. Travel-time to depth conversion**
We convert the radar travel time to depth by iteratively multiplying the velocity of the electromagnetic wave
by the signal's one-way travel time to each IRH. The electromagnetic speed of the radar wave, $v$ (m s$^{-1}$), is
calculated from the dielectric permittivity, $\varepsilon_r$ (dimensionless), and the speed of light in a vacuum, $c$ (3*10$^8$ m
s$^{-1}$), from
$v = \frac{c}{\sqrt{\varepsilon_r}}$   **(1).**



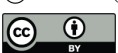

In turn, we calculate the dielectric permittivity for each radar trace from the density, $\rho$ (g cm$^{-3}$), of snow and
ice at depth, as shown in Figure 4, for each radar trace at every range bin (following Kovacs et al., 1995) by
$\epsilon_r = (1.0 + 0.845 * \rho)^2$    **(2).**
We calculate the depth of each subsequent radar sample for each trace in the profile using the radar travel
time and velocity profile from equations 1 and 2, following Hawley et al. (2014) and Lewis et al. (2017).

### 2.5. Internal reflecting horizons

We manually select 10 clear, strong IRHs spaced approximately 5 years apart to consistently trace from
Raven/Dye-2 to Summit Station and throughout the 2017 main traverse (Figure 2). We trace each layer
manually by visually identifying strong amplitude peaks throughout the radargram, starting with the 2016
layer and working downwards. We use a spline interpolation between manual picks to trace each layer along
large amplitude reflections every ~500 – 700 m along the traverse. When a layer appears to bifurcate due to
changes in accumulation, we continue to trace the layer based on the trajectory of surrounding IRHs.
Horizons are not traced in areas where the attenuated signal makes them too difficult to interpret (Figure 3).
We trace layers for each spur starting at the depth of each layer at the corresponding firn core location. We
can trace layers below the depth of some firn cores by tracing them from cores that are deeper or have lower
accumulation rates.

We trace layers between cores using a connect-the-dots approach using the depth-age scale at each firn core.
We trace layers from one firn core to the next before checking that we intersect that core location at the
proper depth for the age of our traced IRH. Note that the depths of several layers at Cores 2 – 16 are located
below the bottom depth of those cores. Since these layers are isochronous, they are used to calculate
accumulation over appropriate time epochs by using dates obtained from intersections with other cores (see
Figure 3).

### 2.6. Accumulation calculations and uncertainty

Finally, we calculate snow accumulation using the firn core depth-age scales, measured and interpolated
depth-density profiles (Figure 4), and traced IRHs (Figure 2). We calculate the water equivalent accumulation
$\dot{b}$ (m w.e. a$^{-1}$) between adjacent IRHs from the depth $z$ (m) and age $t$ (year) of each layer, the average density
$\rho$ (kg m$^{-3}$) between layers, and the density of water $\rho_w$ (1000 kg m$^{-3}$):
$\dot{b} = \frac{1}{t_2-t_1} \int_{z_1}^{z_2} \frac{\rho(z)}{\rho_w} \partial z$    **(3).**




We correct for layer thinning using a Nye (1963) model. The thinning factor has an average value of 0.9993
± 0.0003 and is multiplied by the accumulation rate for each radar trace. For each radar trace, the thinning
factor, $\lambda(z)$, is calculated from the average accumulation $\dot{b}$ (m w.e. a$^{-1}$) of each epoch, average age of the
epoch $a$ (year), and water equivalent thickness of the GrIS $H$ (m), from Morlighem et al. (2014):
$\lambda(z) = e^{-\frac{\dot{b}}{H}a}$        **(4).**

Accumulation uncertainty can arise from independent errors in tracing IRHs, errors from incorrectly dating
firn cores, and/or errors in the densities used for converting from separation distance to water equivalent
accumulation. To reduce tracing errors, we retraced each IRH along the two main traverse paths four times
each. Close inspection of the IRHs reveals that the peaks defining IRHs are within ± 2 radar samples (within
at most ±0.12 m), and incorrectly jumping to the next IRH would result in an error of at most ± 10 samples
(within ±0.55 m). Our average epoch between IRHs is 5.0 years from the firn core depth-age scales, which
corresponds to a maximum tracing error of ~±0.11 m a$^{-1}$ for each epoch, or a maximum error of ±0.061 m
w.e a$^{-1}$, for an average density of 0.55 g cm$^{3}$ over this dataset.

We perform a leave-one-out cross validation to calculate accumulation errors at locations where we do not
have firn core density profiles. Here we choose one of the sixteen firn cores, in addition to the Raven/Dye-2
and Summit cores, to omit from our density interpolation (Figure 4), so that we interpolate density profiles
between adjacent firn cores and a Herron-Langway profile at the missing core location. We find maximum
single-epoch errors of 0.079 m w.e. a$^{-1}$ and maximum RMS (1971 – 2016) errors of 0.046 m w.e. a$^{-1}$ (Table
1) at the location of missing cores, corresponding to 20.1% of the accumulation at that location. These
differences are approximately twice as large at Cores 1 – 6 than Cores 7 – 16 due to larger differences between
measured and interpolated density profiles, likely a result of meltwater percolation and ice lenses (Graeter et
al., 2018).

Similarly, we perform a leave-out-out validation by omitting a firn core density profile location entirely and
interpolating density profiles over a larger distance (e.g. between Core 1 and Core 3). In this case we find
maximum single-epoch errors of 0.057 m w.e. a$^{-1}$ and maximum RMS (1971 – 2016) errors of 0.033 m w.e.
a$^{-1}$.

We conservatively take our accumulation error from missing density measurements to be 0.079 m w.e. a$^{-1}$.
This error highlights the importance of our firn core spacing between 40 – 100 km along the traverse and



confirms that the accuracy of future remotely sensed radar accumulation (e.g. IceBridge snow and
accumulation radars) estimates depend on precise field-based *in situ* density profiles for accurate
accumulation history in the percolation zone. Overly et al. (2016) calculated accumulation in the dry snow
zone using Herron-Langway profiles within 3.5% of accumulation calculated using neutron-probe density
profiles. However, here we show that *in situ* measurements, or accurate meltwater percolation modeling
(Meyer and Hewitt, 2017), are required to correctly calculate SMB in the percolation zone.

**Table 1. Difference between accumulation rates at each GreenTrACS core site calculated using Herron-Langway profiles and firn core**
**density information.**

| Core | RMS average difference ($\mathrm{m\ w.e.\ a^{-1}}$) | Max epoch difference ($\mathrm{m\ w.e.\ a^{-1}}$) | Max Epoch difference (% of acc.) |
|---|---|---|---|
| 1 | 0.046 | 0.079 | 20.1 |
| 2 | 0.025 | 0.061 | 16.2 |
| 3 | 0.037 | 0.074 | 19.9 |
| 4 | 0.028 | 0.045 | 10.7 |
| 5 | 0.026 | 0.054 | 11.5 |
| 6 | 0.038 | 0.052 | 10.0 |
| 7 | 0.015 | 0.026 | 5.4 |
| 8 | 0.026 | 0.045 | 10.3 |
| 9 | 0.030 | 0.049 | 10.9 |
| 10 | 0.019 | 0.039 | 8.5 |
| 11 | 0.023 | 0.035 | 5.0 |
| 12 | 0.018 | 0.027 | 8.2 |
| 13 | 0.025 | 0.031 | 10.7 |
| 14 | 0.019 | 0.027 | 8.2 |
| 15 | 0.010 | 0.016 | 5.3 |
| 16 | 0.014 | 0.025 | 8.2 |


We assume uncertainty in dating the firn cores from annual layer counting to be ±0.5 years (Buchardt et al.,
2012). At the lowest accumulation locations, the smallest distance between layers is 0.15 m w.e. over an
epoch of 4.91 years. This gives an uncertainty in accumulation due to dating of at most ~±0.03 m w.e. $\mathrm{a^{-1}}$.
The error associated with measuring *in situ* firn density has been estimated to be 1.4% (Karlöf et al., 2005).
However, following Hawley et al. (2014) and Lewis et al. (2017), we conservatively assume that our
measurements have a density measurement error of up to twice this large, corresponding to a maximum
accumulation error of ±0.014 m w.e. $\mathrm{a^{-1}}$.

We calculate the total uncertainty from formal error propagation (following Bevington and Robinson, 1992)
from the average accumulation rate $\dot{b}$ = 0.385 m w.e. $\mathrm{a^{-1}}$, average thickness between IRHs $\Delta h$ = 3.56,





uncertainty in tracing $\delta h$, average firn density $\rho$, uncertainty in density measurements $\delta \rho$, average time
period between IRHs $\Delta t$, and uncertainty in core dating $\delta t$. We find a total accumulation rate uncertainty of
0.0709 m w.e. a$^{-1}$ from equation 5.
$$\sigma_b = \sqrt{b^2 \left( \left( \frac{\delta h}{\Delta h} \right)^2 + \left( \frac{\delta t}{\Delta t} \right)^2 + \left( \frac{\delta \rho}{\rho} \right)^2 \right)} \qquad (5)$$

Due to the random and non-systematic nature of these errors, we can assume that they are unlikely to
contribute to a regional or temporal accumulation bias. To calculate uncertainty for accumulation averaged
over multiple epochs ($\sigma_{n-epochs}$) we divide our uncertainty $\sigma_{epoch}$ by the square root of the number of traced
layers (n) at that location.
$$\sigma_{n-epochs} = \frac{\sigma_{epoch}}{\sqrt{n}} \qquad (6).$$

## 2.7. Model comparison

We compare our GreenTrACS accumulation results with annual outputs from Box et al. (2013; hereafter
"Box13"; 1840 – 1999), the Fifth Generation Mesoscale Model (Polar MM5; 1958 – 2008; Burgess et al.,
2010), MAR (1948 – 2015; Fettweis et al., 2016), and the Regional Atmospheric Climate Model (RACMO2;
1958 – 2015; Noël et al., 2018) over common time periods. Grid cell sizes for these model outputs are 5 km,
3 km, 5 km, and 1 km, respectively. For each radar trace we calculate statistically significant differences (at
$\alpha = 0.05$) using a two sample t-test with the GreenTrACS accumulation records for each epoch and RCM
accumulation for each common year. Additionally, we compare our GreenTrACS accumulation with an
accumulation map kriged from 295 firn cores and 20 coastal weather stations (Bales et al., 2009; hereafter
"Bales09"). We perform the same two sample t-test with the reported Bales09 uncertainty of 0.092 m w.e. a$^{-}$
$^{1}$ (Bales et al., 2009).

## 2.8. Accumulation trends

To investigate recent changes in GrIS accumulation, we calculate trends in accumulation across our GPR and
GreenTrACS firn core dataset. We fit a linear model to the accumulation time series for each radar trace and
analyze the trend for both slope and statistical significance. Likewise, we calculate trends and their statistical
significance for total precipitation (snowfall + rainfall) for MAR and RACMO2 grid cells from 1996 through
the end of both models' temporal coverage. We can compare these results with our accumulation trends since
precipitation and accumulation are nearly identical above the equilibrium line altitude, due to zero runoff and
negligible sublimation within the percolation zone.



  **2.9. Storm track changes**

To investigate the potential role of changing storm tracks in precipitation changes over the Western GrIS, we
utilize the updated Serreze (2009) storm track database. This database contains six-hour interval positions of
extratropical cyclone storm centers on a 2.5° grid. These centers are defined when a gridpoint sea level
pressure is surrounded by gridpoints at least 2 mb higher than the central point (Serreze, 2009). We calculate
the total number of days in which a storm center is located within our region of interest for each season. To
determine statistical significance, we run a two sample t-test on the number of storms in our region of interest
between 1958 – 1996 compared with 1996 – 2016.
**3.  Results and discussion**
**3.1. Firn core and GPR accumulation records**
Figure 1 displays the mean accumulation at each location along the traverse route, with higher accumulation
rates in the southwest and lower accumulation rates at higher elevations of the ice sheet interior, broadly
consistent with previously published accumulation compilations (e.g. Bales et al., 2009) and RCM output
(Box et al., 2013; Burgess et al., 2010; Fettweis et al., 2016; Noël et al., 2018). We analyze localized
differences between GPR derived accumulation and these RCMs in Section 3.3. There is an especially high
accumulation zone near Core 11 (0.685 m w.e. $a^{-1}$), nearly double the accumulation at Core 10 (0.453 m w.e.
$a^{-1}$) and Core 12 (0.327 m w.e. $a^{-1}$), respectively situated only 43 km northwest and 73 km southwest of Core
11. In the GPR data, the number of traceable IRHs is highest towards the interior of the ice sheet and lowest
in warmer areas towards the coast and in the south, where refrozen percolated melt water from enhanced
surface melt attenuates the radar signal and reduces the number of observable IRHs (Brown et al., 2011;
Figure 3).
**3.2. Validation with past measurements**
We validate our accumulation record with published core records from the PARCA campaign and
accumulation data from the NASA IceBridge program. The locations of GreenTrACS Core sites 2, 5, 9, 10,
11, 14, 15, and 16 were chosen to reoccupy PARCA core locations 6745, 6945, 7147, 7247, 7249, NASA-
U, 7347, and 7345, respectively. These GreenTrACS cores overlap with the accumulation history of each
PARCA core and extend the record from 1997/1998 to 2016/2017. Accumulation rates derived from
GreenTrACS firn cores are within error of those determined from corresponding PARCA cores during the
period of overlap. Figure 5 compares the accumulation records from PARCA sites 6745, 6945, 7345, and
NASA-U to their corresponding GreenTrACS cores, demonstrating that each pair of cores has similar long-
term mean accumulation and nearly identical decadal variability. Thus, we have confidence in firn core





derived accumulation rates that are used in subsequent GPR calculations of accumulation rates throughout
the GreenTrACS traverse.

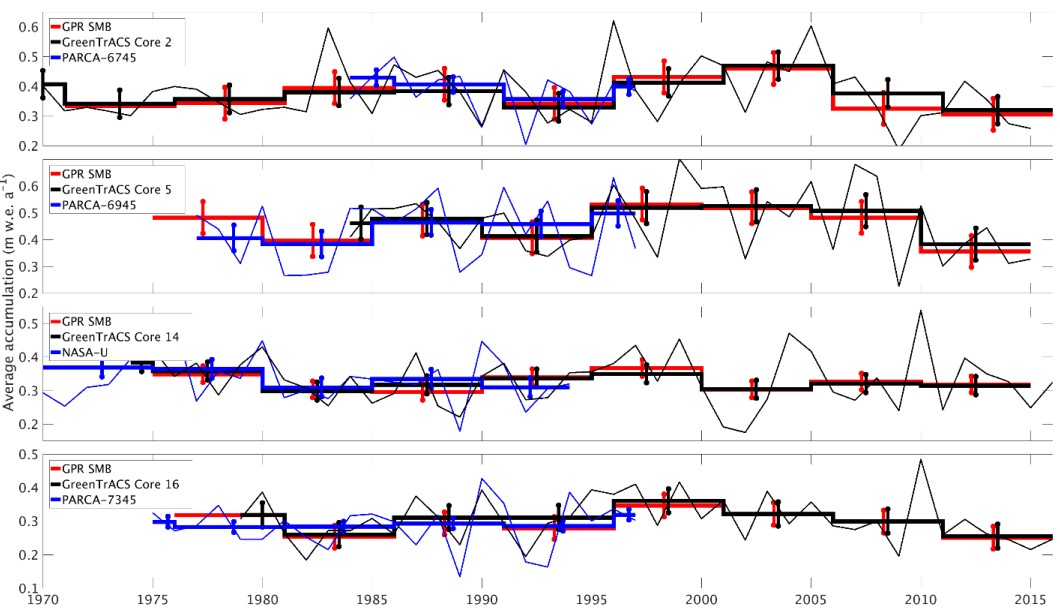


**Figure 5. Accumulation from GPR and collected firn cores (this study) compared with cores from the PARCA Campaign. Thin lines**
**represent annual PARCA (blue) and GreenTrACS (black) firn core accumulation, while thick lines are 5-year averages over**
**corresponding GPR epochs. Error bars represent one standard deviation over each epoch. GPR and PARCA accumulation averages**
**and decadal trends are statistically indistinguishable.**

Average (1966 – 2016) GPR accumulation is statistically indistinguishable with average (1962 – 2014)
IceBridge Accumulation Radar measurements analyzed by Lewis et al. (2017), with an RMS difference of
$0.0387 \pm 0.0327$ m w.e. a$^{-1}$ along a total of 562.5 km of overlap (Figure 6). The disagreement is largest at
lower elevations, where Herron and Langway (1980) density profiles used in Lewis et al. (2017) differ the
most from GreenTrACS firn core density profiles in the upper 30 m of firn, demonstrating the importance of
field observations for calibration and validation. The close agreement at higher elevations is illustrated in
Figure 7a, where our GreenTrACS accumulation measurements are statistically indistinguishable from the
IceBridge radar-derived accumulation (Lewis et al., 2017) along the 285 km A – A' transect on Figure 1.
Notice that the uncertainty in GreenTrACS accumulation progressively decreases higher in the percolation
zone and into the dry snow zone (towards the right in Figure 7) along this transect as density becomes less
heterogeneous from fewer melt layers (Graeter et al., 2018) and IRHs become easier to trace.






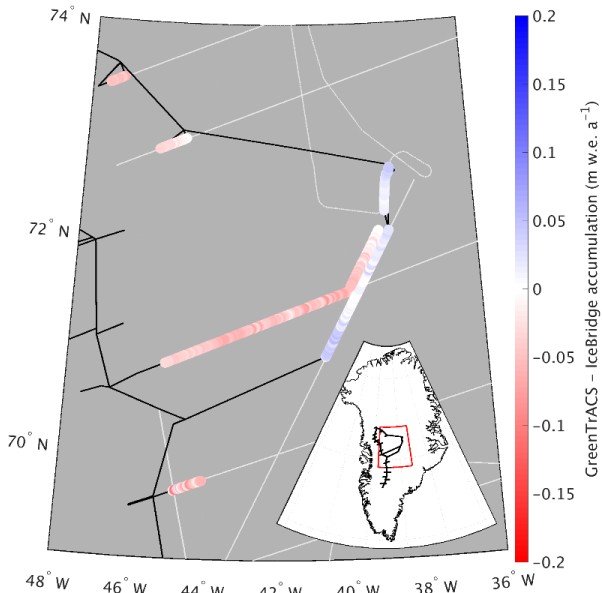

**Figure 6. Difference between averaged (1966 – 2016) GreenTrACS accumulation and average (1962 – 2014) IceBridge Accumulation Radar rates from Lewis et al. (2017) across all 562.5 km of overlap. Spatially overlapping section of 2016 and 2017 traverses displayed as adjacent tracks. Also showing extent of GreenTrACS traverse (black) and IceBridge accumulation radar (grey). Inset shows map location with respect to GreenTrACS traverse (black).**

### 3.3. Comparison to modelled accumulation

We assess differences between RCM accumulation output and GreenTrACS accumulation record at each firn core site, two of which are shown in Figure 8. In general, year-to-year correlations between GreenTrACS firn core accumulation records and RCM output for the corresponding grid cell are strong, positive, and statistically significant (Table 2). On average, GreenTrACS firn cores' correlation coefficient with MAR output is 0.718, with PolarMM5 is 0.701, with Box13 is 0.607, and with RACMO2 is 0.763. Every correlation is statistically significant at $p < 0.05$ except for Cores 7 and 11 with Box13. We do not report a correlation coefficient for Core 11 and Box13 because they only share two common years. Temporal correlation coefficients remain high even at locations with large magnitude differences between RCM output and firn core accumulation. For example, the Box13 model overestimates accumulation at Core 15 by $0.15 \pm 0.05$ m w.e. a$^{-1}$, on average, but the model output has a correlation coefficient of 0.48 with Core 15 (Table 2) and matches years of high accumulation (e.g. 1987, 1990, and 1996) and low accumulation (e.g. 1981, 1989, 1992).





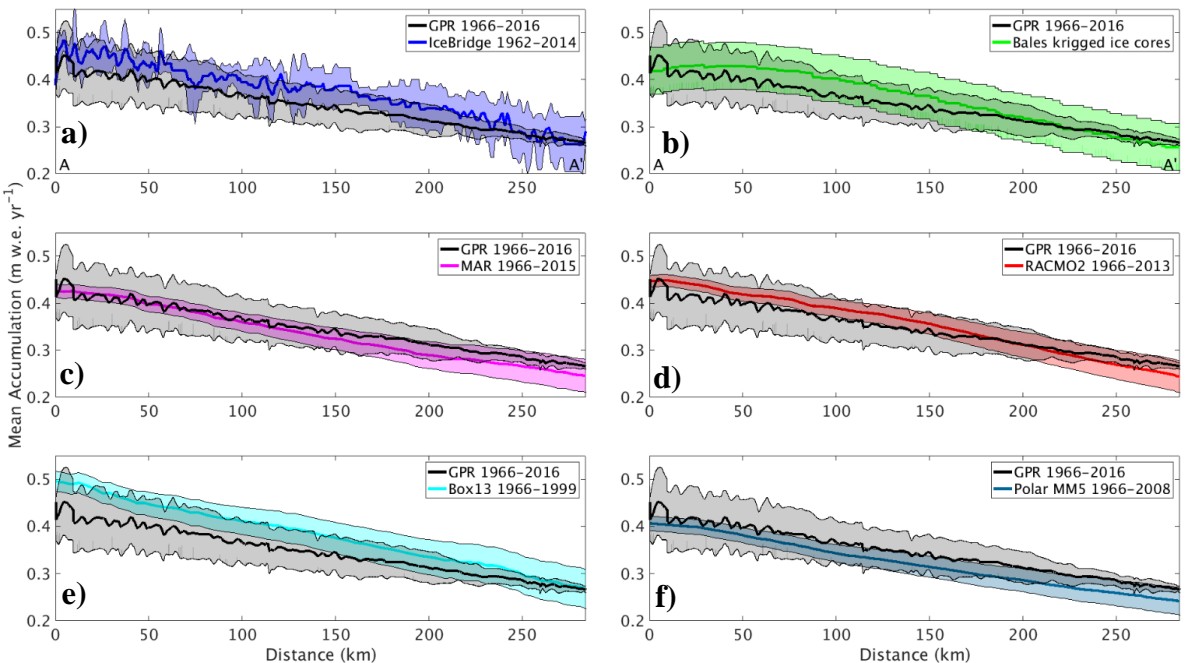

Figure 7. Average GreenTrACS GPR accumulation (black) compared with a) IceBridge accumulation radar, b) Bales09 krigged ice core map, c) MAR, d) RACMO2, e) Box13, and f) Polar MM5. GPR measurements are statistically indistinguishable from each of the other measurements along this 285 km transect in the dry snow zone (A – A' on Figure 1).

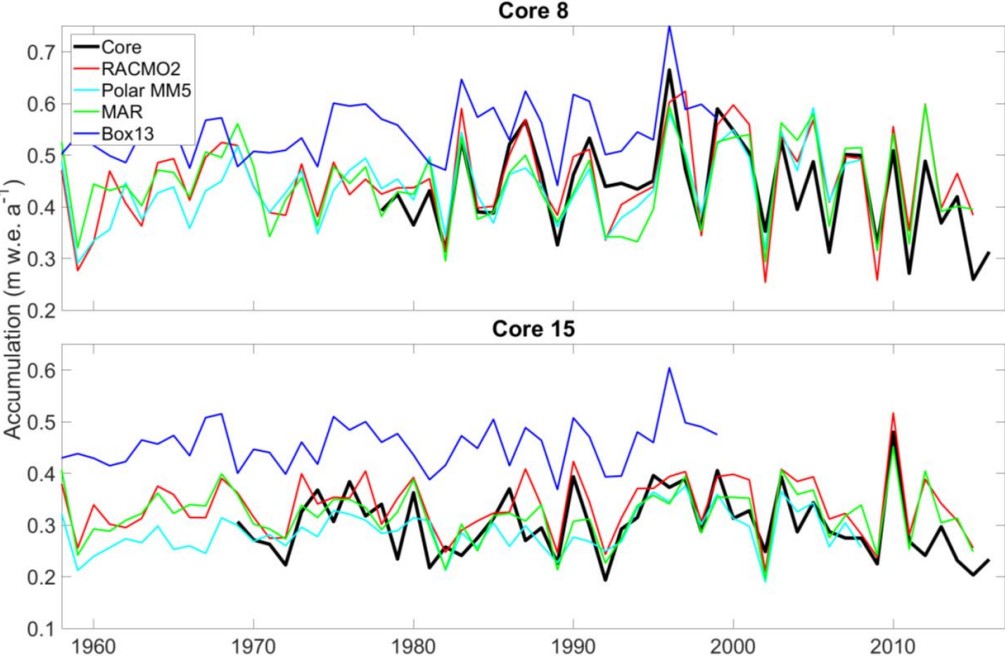

Figure 8. Accumulation record at GreenTrACS Core 8 and Core 15 (black) compared with RCM output from RACMO2 (red), Polar MM5 (cyan), MAR (green), and Box13 (blue). We find statistically significant Pearson correlation coefficients between GreenTrACS and RCM accumulation rates for these cores (see Table 2).




**Table 2. Pearson correlation coefficients between accumulation rate time series from firn cores and co-located RCM output over their**
**common time period[#].**

| | Available data period | MAR | PolarMM5 | Box13 | RACMO2 |
|---|---|---|---|---|---|
| Core1 | 1966 – 2016 | **0.70** | **0.66** | **0.56** | **0.73** |
| Core2 | 1969 – 2016 | **0.75** | **0.77** | **0.62** | **0.79** |
| Core3 | 1971 – 2016 | **0.72** | **0.69** | **0.63** | **0.74** |
| Core4 | 1977 – 2016 | **0.79** | **0.74** | **0.72** | **0.72** |
| Core5 | 1984 – 2016 | **0.81** | **0.80** | **0.60** | **0.79** |
| Core6 | 1985 – 2016 | **0.76** | **0.76** | **0.65** | **0.83** |
| Core7 | 1993 – 2016 | **0.81** | **0.82** | 0.61 | **0.73** |
| Core8 | 1978 – 2017 | **0.78** | **0.77** | **0.69** | **0.81** |
| Core9 | 1984 – 2017 | **0.68** | **0.75** | **0.74** | **0.79** |
| Core10 | 1984 – 2017 | **0.88** | **0.80** | **0.80** | **0.80** |
| Core11 | 1997 – 2017 | **0.75** | **0.59** | N/A | **0.75** |
| Core12 | 1962 – 2017 | **0.6** | **0.54** | **0.53** | **0.64** |
| Core13 | 1955 – 2017 | **0.51** | **0.62** | **0.37** | **0.76** |
| Core14 | 1974 – 2017 | **0.70** | **0.62** | **0.46** | **0.74** |
| Core15 | 1969 – 2017 | **0.68** | **0.63** | **0.48** | **0.75** |
| Core16 | 1979 – 2017 | **0.79** | **0.77** | **0.66** | **0.88** |

[#]Statistically significant correlations ($p < 0.05$) are bold

We also assess spatial differences between GreenTrACS accumulation and mean RCM accumulation
averaged over several decades. Figure 9 shows that differences between GreenTrACS accumulation and
RCM output increase in magnitude, become more spatially heterogeneous, and vary by model at lower
elevations of the ice sheet where topographic variations are larger and surface melt increases. Averaged over
all 4436 km of the traverse, the RMS difference between each model and GreenTrACS accumulation is 0.068
± 0.065 (MAR), 0.0562 ± 0.0548 (RACMO2), 0.0822 ± 0.0702 (Box13), 0.048 ± 0.045 (Polar MM5), and
0.0475 ± 0.0445 m w.e. a$^{-1}$ (Bales09). We find that RCM differences from GreenTrACS accumulation are
small in the dry snow zone (Figure 9). For example, Figure 7 shows that average GreenTrACS accumulation
measurements from 1966 – 2016 along the A – A' transect in Figure 1 are statistically indistinguishable from
those derived from the Bales09 krigged ice core map (Figure 7b), MAR (1966 – 2015; Figure 7c), RACMO2
(1966 – 2013; Figure 7d), Box13 (1966 – 1999; Figure 7e), and Polar MM5 (1966 – 2008; Figure 7f).

However, the high spatial resolution of our dataset shows significant accumulation variability not captured
in model output (Figure 9). For example, Polar MM5 and MAR both underestimate accumulation between
Core 4 and Summit, while overestimating accumulation to the west of Cores 10 – 12. Likewise, RACMO2
overestimates accumulation between Raven/Dye-2 and Core 5 by 0.03 to 0.08 m w.e. a$^{-1}$ and shows
statistically significant differences east of Cores 11 and 12. Bales09 accurately calculates accumulation along
most of the 2016 traverse, but overestimates accumulation west of Cores 11 and 12 by 0.135 ± 0.041 m w.e.





a⁻¹. Finally, Box13 overestimates accumulation along many of the western spurs and has statistically
significant overestimations of 0.1 to 0.4 m w.e. a⁻¹ between Cores 10 and 16. Box13 overestimates 67.8% of
the data in the Core 10 – 16 region by at least 0.1 m w.e. a⁻¹, and 6.6% of that data by at least 0.2 m w.e. a⁻¹.

Our study is almost entirely contained within drainage basin E from Vernon et al. (2013), who note that basin
E is the only major Greenland drainage basin with no statistically significant differences in SMB between
the four RCMs. However, differences of 0.1 to 0.4 m w.e. a⁻¹ exist when we look at a local (sub-drainage-
basin) scale for each model. All four of the RCMs overestimate accumulation along the western spur of Core
11 and they all underestimate accumulation along the eastern spur of Core 5 (Figure 9).






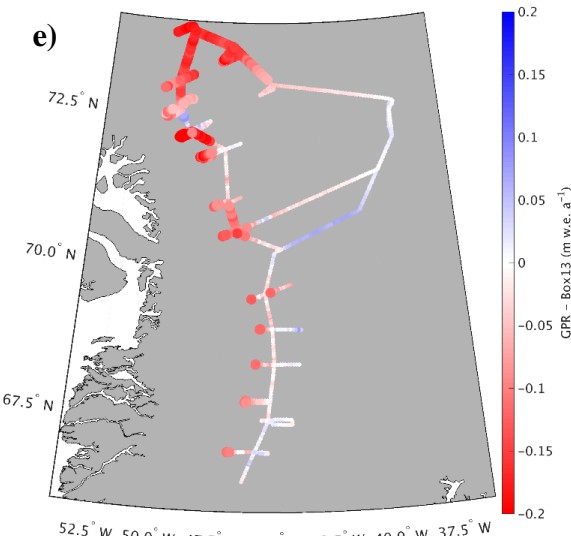


Figure 9. Differences between GreenTrACS accumulation and a) Polar MM5, b) MAR, c) Bales09, d) RACMO2, and e) Box13 accumulation averaged over the corresponding time periods. Large dots show statistically significant differences from GreenTrACS accumulation.


In summary, the RCMs do an excellent job of calculating accumulation averaged over this drainage basin, with RMS values between 0.048 and 0.0822 m w.e. a$^{-1}$, but there are larger differences of 0.1 to 0.4 m w.e. a$^{-1}$ between model and GPR accumulation on local scales. Differences between GreenTrACS and RCM accumulation are largest in areas concurrent with the fewest, shortest, and/or most outdated *in situ* measurements. For example, the GPR vs. model differences near Cores 11, 12, and 13 are relatively large for all RCMs, despite Core 11 being co-located with PARCA 7249. However, the PARCA cores were collected over 20 years ago, and Core 11 only spanned 7 years because of the high accumulation rate at that site. This highlights the importance of collecting updated field-based measurements to calibrate remotely sensed data and RCM output.

519

### 3.4. Accumulation temporal trends

In most locations, there are no statistically significant trends in the GreenTrACS accumulation record from 1966 through the mid-1990s. However, a changepoint analysis (Lavielle, 2005) reveals that accumulation in the Western GrIS percolation zone changed significantly after the 1995 – 1996 accumulation year. Since 1996, our record indicates a statistically significant average accumulation decrease of 0.009 ± 0.005 m w.e a$^{-2}$, or 2.4 ± 1.5 % a$^{-1}$, from 1996 to 2017. Although we observe fewer statistically significant accumulation




trends when we expand this analysis to include the entire temporal duration for each firn core, the sign of the
trend at each core site does not change.

In Figure 10, we compare the negative accumulation trend in the GreenTrACS record (1996 – 2016) to best-
fit linear trends in total precipitation (rain + snowfall) across the ice sheet in MAR and RACMO2 simulations
over the 1996 – 2015 and 1996 – 2013 periods, respectively. Also shown in Figure 10 are 1996 – 2016
accumulation trends for all 16 GreenTrACS firn cores (squares), accumulation trends from ACT10A (1996
– 2010), ACT10B (1996 – 2010), ACT10C (1996 – 2010), D4 (1991 – 2002), D5 (1991 – 2002), Katie (1991
– 2002), Sandy (1991 – 2002), and Summit 2010 (1991 – 2010) ice/firn cores (stars on ice sheet), and
precipitation trends from coastal weather stations (Mernild et al., 2014; stars on coast). Statistically
significant trends ($p < 0.05$) in core data are indicated by black dots, while statistically significant trends in
the MAR and RACMO2 output are stippled in black.

We find strong agreement between the accumulation decrease in the GreenTrACS record and widespread
precipitation decreases in the RCMs over the study area (Figure 10). On average, the RCMs have a more
negative precipitation trend than the GreenTrACS record by $0.003 \pm 0.005$ for MAR and $0.0016 \pm 0.0051$ m
w.e. $a^{-2}$ for RACMO2. Vernon et al. (2013) show a melt-driven decrease in SMB across this drainage basin
of 31.1% (ECMWFd), 61.6% (RACMO2), 76.5% (MAR), and 33.5% (Polar MM5) for the 1996 – 2008
period. The negative precipitation trends of $2.4 \pm 1.5$ % $a^{-1}$ (Figure 10d) indicate a total of 2539.4 fewer Gt
of precipitation and a total of 5159.1 additional Gt of melt (not shown) over 1996 – 2013 across the GrIS.
Thus, our analysis suggests that a significant decline in snow accumulation contributes to declining SMB
throughout the Western GrIS over recent decades, in addition to increasing surface melt from rising
temperatures (van den Broeke et al., 2009, 2016).





**Figure 10. Best fit linear trends for each grid cell showing magnitude (left) and percent (right) changes in total precipitation for a) and b) MAR (1996 – 2015) and c) and d) RACMO2 (1996 – 2013). Statistically significant RCM grid cell trends are stippled black. Also shown are accumulation trends for GreenTrACS firn cores (squares), ACT10A, ACT10B, ACT10C, D4, D5, Katie, Sandy, Summit 2010, and Raven/Dye-2 cores (stars on ice sheet) and precipitation trends from Mernild et al (2014; stars on coast) with statistically significant trends indicated by black dots.**


## 3.5. Effects of melt on accumulation trends

Increased melt throughout the 1996 – 2016 period is a confounding variable when analyzing trends in
accumulation. With increased melt over the past several decades in this region, meltwater percolates down
through several years of firn (Benson, 1962; Graeter et al., 2018; Harper et al., 2012; Wong et al., 2013).



This movement of mass into lower years can artificially increase the mass balance at depth and lower the
mass balance during the most recent years, which have not experienced as much meltwater percolation from
more recent annual layers. Therefore, it is necessary to evaluate the degree to which the recent accumulation
decrease in the GreenTrACS record is biased by the recent increase in surface melt and percolation.

Figure 11 compares the 1996 – 2016 mass balance trends with 1996 – 2016 average melt for each of the
sixteen GreenTrACS firn cores. If we exclude Core 11 (which only dates back to 1997 and has a highly
negative SMB trend), the linear regression is statistically significant with $p = 0.04$ (Figure 11). Note that both
the measured Core 11 SMB trend and RCM trends at that location are so negative, with that small amount of
average melt, that the linear trend is no longer significant if that point is included in the calculations. On
average, we find larger negative accumulation trends ($-7 \times 10^{-3}$ to $-10 \times 10^{-3}$ m w.e. $a^{-2}$) at the lower latitude
cores that experience more melt, supporting the hypothesis that meltwater percolation and refreezing are
enhancing the negative accumulation trend.

However, several other lines of evidence support a negative accumulation trend in the study area since 1996.
First, we find statistically significant negative accumulation trends at Cores 10, 11, 12, 13, 15, and 16, each
of which experience $<1.6$ cm $a^{-1}$ of meltwater percolation on average (Figure 11). Additionally, we have
confidence that GreenTrACS accumulation trends reported here are not artifacts of meltwater percolation
because both MAR and RACMO2 have similar trends in precipitation (Figure 10). Finally, we evaluate the
maximum effect meltwater percolation could have on GreenTrACS accumulation trends over 1996 – 2016.
The largest melt layer from our sixteen ice cores occurred during 2003 – 2004 in Core 1 and contains 0.364
m of ice, equivalent to 0.333 m w.e. (Graeter et al., 2018). We add this percolation to nine years' of
accumulation using a sine wave (percolation magnitude 0, 0.5, 1, 0.5, 0, -0.5, -1, -0.5, 0), square wave (0, 0,
0, 1, 1, 1, 0, 0, 0), and triangle wave (0, 0.25, 0.5, 0.75, 1, 0.75, 0.5, 0.25, 0) weighted kernel, before re-
computing hypothetical accumulation trends over the same time period with additional meltwater
percolation. Regardless of the wave-type choice, re-calculated trends remain within a factor of two of the
original SMB trends and do not change sign with additional percolation.



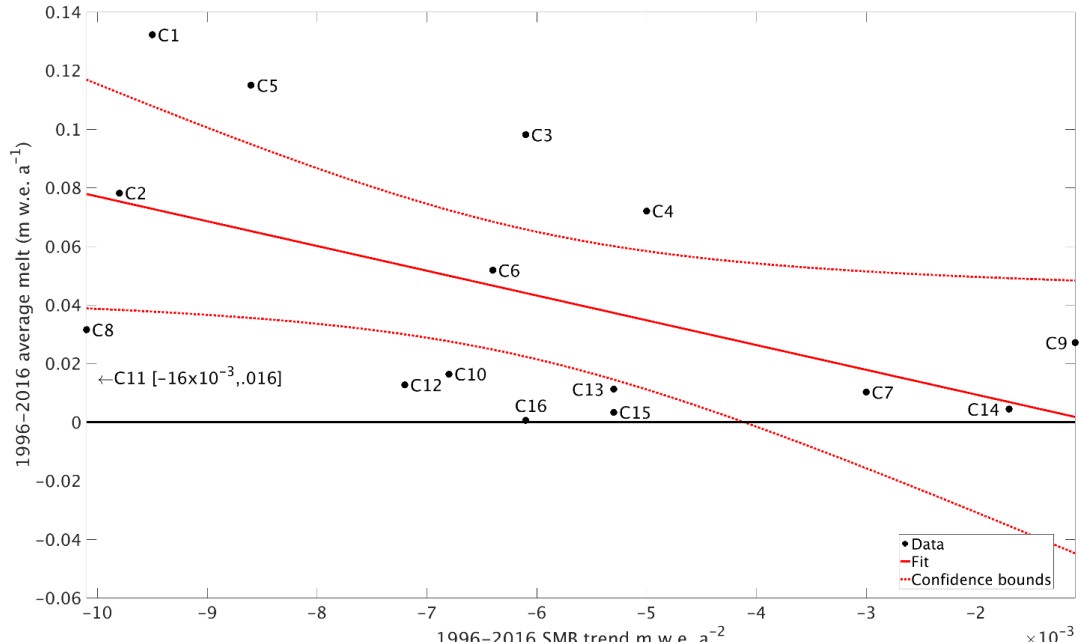

**Figure 11. Relationship between 1996 – 2016 SMB trend and 1996 – 2016 melt for each of the 16 GreenTrACS firn cores (black circles). Red line shows linear best fit, dotted line shows 95% confidence boundary.**

### 3.6. Atmospheric circulation drivers of the recent accumulation decline

Our analysis indicates that snow accumulation has been declining in Western Greenland since 1996, despite significant warming and resulting increases in saturation vapor pressure from the Clausius-Clapeyron relationship. Instead, precipitation decreases over Western Greenland likely result from changes in atmospheric and/or oceanic circulation. Mernild et al. (2014) and Auger et al. (2017) found that the positive phase of the Atlantic Multidecadal Oscillation (AMO) is associated with a precipitation increase over interior and Southwestern Greenland based on ice core records and the Japanese Meteorological Agency 55 Year Reanalysis (JRA-55; Kobayashi et al., 2015), respectively. In direct contrast with these findings, the decline in Western Greenland accumulation documented in the GreenTrACS record began in the mid-1990s, contemporaneous with a switch to the AMO positive phase.

We hypothesize that the differences between our results and those of Auger et al. (2017) and Mernild et al. (2014) stem from different causes. Auger et al. (2017) validated the reanalysis data by demonstrating that JRA-55 precipitation at Nuuk, Greenland is significantly correlated with Nuuk station data from 1958 – 2013. Furthermore, coastal precipitation in Western Greenland is strongly and significantly ($p < 0.05$) correlated with precipitation over the interior Western GrIS in the JRA-55 dataset (not shown). However, Mernild et al. (2014) found that coastal Greenland precipitation is anti-correlated with ice core accumulation records from





the interior GrIS from 1900 to 2000. This suggests that JRA-55 precipitation data, which is not constrained
by ice core accumulation records, should be interpreted with caution over the interior GrIS. Mernild et al.
(2014) concluded that positive AMO conditions favor higher precipitation over the interior GrIS based on
the previous positive AMO phase (1920s to mid-1960s), contrasting with lower accumulation during the
negative AMO phases (mid-1960s to mid-1990s and prior to the 1920s). However, Mernild et al. (2014) state
that the ice core composite record in their analysis may be biased from 1995 – 2000, and they do not analyze
precipitation trends after 2000. Thus, the decline in Western GrIS accumulation documented in the
GreenTrACS cores during the latest positive AMO phase from 1996 to 2017 was not captured in the Mernild
et al. (2014) analysis. Our results suggest that factors other than the AMO are behind the decline in Western
GrIS accumulation since 1996.

We find that the decrease in accumulation over the Western GrIS is associated with a significant decrease in
the number of storm-days since 1996. The GreenTrACS region experienced an average of 115.8 ± 15.3 storm-
days per year over 1958 – 1996 and 96.2 ± 27.3 storm-days per year over 1996 – 2016. A two sample t-test
indicates that this 17% decline in storm-days is statistically significant (p < 0.001). The largest decrease in
storm-days (25%) over the GreenTrACS region occurred during summer, with 56.4 ± 6.1 storm-days per
summer from 1958 – 1996 and 42.3 ± 17.4 storm-days per summer from 1996 – 2016 (p < 0.0001; Figure
12b). We also find an increase in the number of storm-days in the Northwestern GrIS near Thule (not shown).

The decline in summer storm-days indicates a relationship with well-documented stronger summer blocking
over Greenland over the past two decades (Hanna et al., 2013; McLeod and Mote, 2016), with a positive
Greenland Blocking Index during 17 out of 21 summers between 1996 – 2016 (Hanna et al., 2016). The June
– August GBI had a statistically significant positive trend of 1.87 (unitless; normalized to 1951 – 2000) from
1991 – 2015 (Hanna et al., 2016). The summertime 500 mbar geopotential height increased 50 – 70 m over
the 1996 – 2016 period compared with the 1979 – 1996 baseline (Figure 12c), indicating stronger blocking
that we suggest likely reduced precipitation over the central GrIS by deflecting storms poleward from the
Greenland interior. This is consistent with an observed $0.9 \pm 0.3\%$ a$^{-1}$ decrease in JJA cloud cover over
Greenland from 1995-2009, with the largest decreases in the GreenTrACS region (Hofer et al., 2017).
Furthermore, we find a strong negative correlation between ERA-Interim 1979 – 2015 June – August (JJA)
GBI and JJA precipitation in both MAR (Figure 12d) and RACMO2 (not shown) across the central and
southern GrIS. These results suggest that the blocking-induced accumulation decline observed in the
GreenTrACS region is representative of a broader pattern over the GrIS, with the exception of Northwest
Greenland where poleward blocking has increased storm-days (not shown) and accumulation (Figure 12d).






The effect of summertime Greenland blocking has been discussed primarily in the context of increasing
surface melt (Hanna et al.. 2013; Ballinger et al., 2017; Hanna et al., 2018; Hofer et al., 2017), while the
effect of blocking on precipitation has received less attention (Hanna et al., 2013; McLeod and Mote, 2016).
Our results highlight that stronger summer blocking reduces GrIS SMB through both an increase in surface
melting and a decrease in accumulation. Stronger summer blocking has been tied to an observed increase in
surface melt since 1996 across the Western GrIS percolation zone (Graeter et al., 2018), and to the July 2012
melt event, during which 98.6% of the GrIS experienced melt (Nghiem et al., 2012). We show here with *in*
*situ* data that snow accumulation has declined in this same region as strong blocking has decreased the
number of summer storm-days. Presently, none of the GBI outputs from the Coupled Model Intercomparison
Project 5 (CMIP5) suite of global climate models accurately capture the recent summer GBI increase (Hanna
et al., 2018). Improved predictions of summertime Greenland blocking under future anthropogenic forcing
scenarios are therefore critical for accurately predicting Greenland SMB and its contribution to sea level rise.

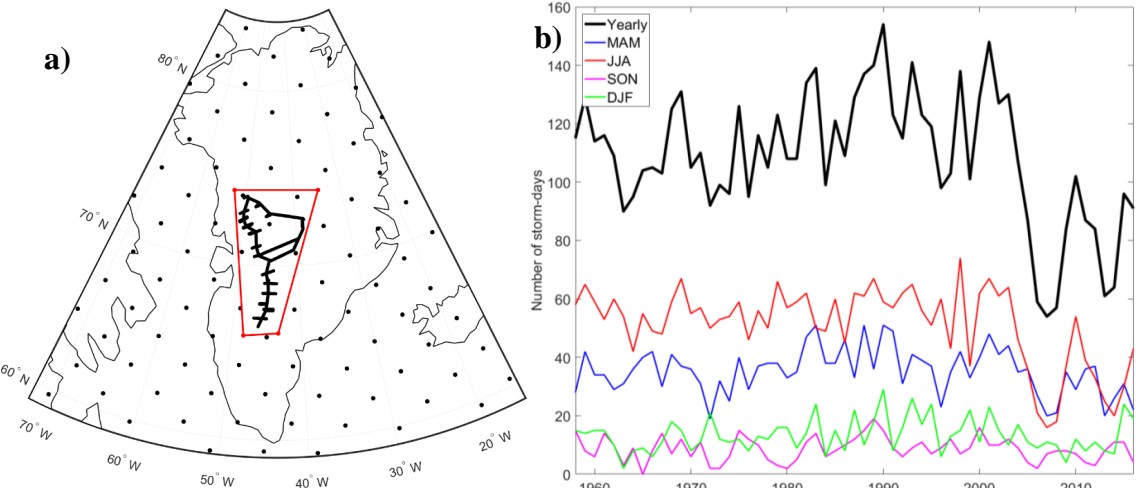




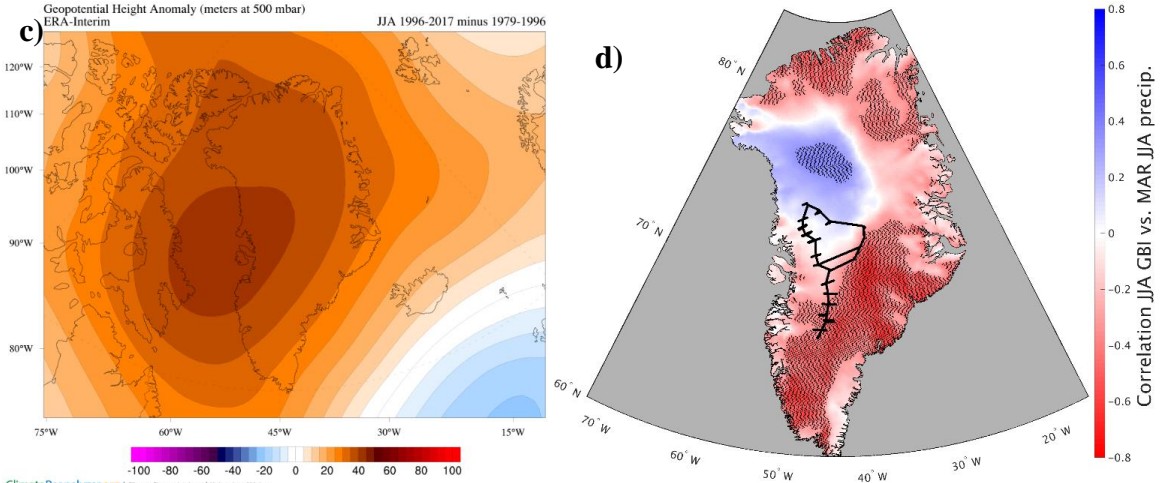



**Figure 12. a) (Serreze, 2009) gridded storm track dataset showing location of GreenTrACS traverse and inquiry box. b) Total number**
**of storm-days within inquiry box for annual and seasonal periods. Horizontal black lines show a decrease in 1958 – 1996 to 1996 – 2016**
**average number of storm-days within this region. c) 500 mbar geopotential height change over Greenland showing 1996 – 2016 minus**
**1979 – 1996 for the summer season. Image obtained using Climate Reanalyzer (http://cci-reanalyzer.org), Climate Change Institute,**
**University of Maine, United States. d) Correlation between June – August Greenland Blocking Index and MAR June – August**
**precipitation. Statistically significant RCM grid cell correlations are stippled black. GreenTrACS traverse is shown in black.**
**4. Conclusions**
We have developed a new dataset of accumulation rates over the western interior of the Greenland ice sheet
spanning the past 20 – 60 years, based on sixteen 22 – 32 m long firn cores and 4436 km of *in situ* GPR
accumulation data. This accumulation record is internally consistent across the dataset and is validated by
previous *in situ* field measurements and other radar-derived accumulation measurements (e.g Lewis et al.,

671 2017).


Overall, the Polar MM5, MAR, Box13, and RACMO2 Regional Climate Models accurately capture large
spatial patterns in accumulation over the GrIS, but show statistically significant differences from GPR
accumulation on a regional basis. The average RMS difference between each model and GreenTrACS
accumulation is $0.068 \pm 0.065$ (MAR), $0.048 \pm 0.045$ (Polar MM5), $0.0822 \pm 0.0702$ (Box13), $0.0562 \pm$
$0.0548$ (RACMO2), and $0.0475 \pm 0.0445$ m w.e. $a^{-1}$ (Bales09). These differences are on the same order as
the uncertainties in the GreenTrACS and RCM accumulation estimates. While these average differences are
small, we find differences of 0.1 to 0.4 m w.e. $a^{-1}$ when we investigate at a local scale for each model.

While global climate models predict a 21$^{st}$-century increase in precipitation over the GrIS (e.g. Bintanja and
Selten, 2014), we observe a decrease in precipitation across the Western GrIS from 1996 – 2016 using records





from firn cores, GPR, and published RCMs. We believe this study is the first to identify widespread negative
GrIS precipitation trends during this period of enhanced surface melt, evident in these RCMs and our field
observations (Graeter et al., 2018).

We attribute the decrease in accumulation over the Western GrIS between 1996 and 2016 to more persistently
positive Greenland blocking in the summer. We find a statistically significant 25% reduction in the number
of summer storms that precipitate over the GreenTrACS region since 1996. While warming temperatures
from anthropogenic forcing and enhanced summer blocking have increased melt across the western
percolation zone, here we show that summer blocking has also contributed to declining precipitation over the
past two decades. This has led to a strongly negative SMB trend on both the input and output sides of the
SMB equation that may not be accurately captured in global climate models that are currently unable to
reproduce the recent increase in blocking. This highlights the importance of improving GCM projections of
future summer blocking to accurately forecast Greenland precipitation and melt rates under stronger
greenhouse gas forcing.

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
