# Peer review of "Recent Precipitation Decrease Across the Western Greenland Ice Sheet 1"

_The Cryosphere, 2019_

## Referee Comment (RC1) · Anonymous Referee #1 · 22 Jul 2019

The manuscript "Recent Precipitation Decrease Across the Western Greenland Ice Sheet Percolation Zone" by Lewis et al. presents large scale GPR transects and accumulation derivations thereof for more than 4400km of the Western GrIS. Such data are combined with firn cores to enable layer dating and accumulation calculations from density measurements. Vertical in-situ data allow accumulation derivations for the last 2 to 6 decades enabling trend assessments. In-situ trends are compared with RCM outputs to analyze for changes in accumulation and precipitation in relation with global temperature changes. The authors describe significant decreases in accumulation rates within the last 2 decades, which they attribute to shifting storm tracks reducing precipitation mainly for the summer months and increasing surface melt. I consider the presented work as novel and certainly significant for the scientific community especially because of the extensive data collection presented in this work. However, some redundancies, imprecise descriptions and the confusing structure of the manuscript prevent publication in the current state. I recommend to focus more on conciseness and maybe reconsider the total volume of the presented data. How about splitting into 2 manuscripts: one presenting the in-situ data including validation/ comparison with RCM results and the subsequent dealing with implications and atmospheric circulation simulations. Right now, the reader gets a bit lost in all the error/ uncertainty analyses combined with validation proofs for numerous statistics. Major points of criticism are: âǍć The structure of the manuscript is very confusing. The methods section comprises large fractions of discussion and data interpretation. Please revise the structure and attempt to shorten the manuscript whenever possible. The introduction comprises almost 3 pages. It is clear to me that you want to introduce all relevant literature and topics, which are presented. However, if splitting into 2 manuscripts (see above), you could certainly focus more on less different topics. Parts, which could be shortened are L54ff and L89ff. âǍć At least to me, it remains unclear how specific values are determined. For instance, epoch and annual accumulation values are hard to distinguish. It would be better to clearly distinguish in between these two. Did you actually pick each individual layer in the radar data or just for specific locations where layer resolution is clear or just the 5 year layers as indicated in Fig. 5? This remains unclear, same for the accumulation calculations. You state that 1m fractions as well as 3cm parts of the cores are analyzed (L233ff) in the field and lab. Were those core fragments further cut for more highly resolved density measurements? In addition, average melt rates in Fig. 11 and discussed in Section 3.5 are not adequately explained. I don't see how such values are generated (derived from RCMs, calculated in accordance to observed ice lenses as in L581?). RMS values describing deviations from RCMs lack an explanation for the uncertainty range. In summary, I must admit, I got lost with all the uncertainty values being presented. What are sigma_epoch errors, how are these values related to sigma_accumulation-rate? I recommend to work carefully on the respective sections and maybe include a sketch of the applied workflow to derive accumulation data from

radar IRHs. • Just for clarification: The accumulation rate uncertainty is 71kg/m2/a, I interpret this value as the max accuracy you can achieve from GPR transects The RMS deviation to IceBridge accumulation rates is 39kg/m2/a, which is within the error margins. For annual accumulation rates in Fig. 5, I would expect to have error margins as stated above being included. How reliable is a 5-year standard deviation in accumulation rates? The RMS deviation to RCMs is 48-82kg/m2/a and again within the error margins of the radar. Annual trends in precip are at 7kg/m2/a2. Consequently, you would need at least a 10 year period to reach the error margins for deriving trends, right? Max. single epoch errors of 79kg/m2/a are found for Herron-Langway comparisons with RMS deviations of 46kg/m2/a How is the vertical resolution limit of the 400MHz antenna calculated? For firn of roh_s=550 kg/m3 you would receive a v_mean of 0.2m/ns resulting in a wavelength of 0.5m. Resolution limits are sometimes defined as half of the wavelength or $\frac{1}{4}$*lambda. How do you come up with 0.35m? • You discuss several times errors introduced by percolating melt water. Heilig et al. (2018) measured the seasonal mass flux from snow into underlying firn at Raven to be at >50kg/m2 (in your preferred units >0.05m w.e.) for summer 2016. Can you clearly date back ice lenses or is the mentioned ice lens from 2003/04 a result of several melt seasons? What about summer 2012? Shouldn't there be a thicker ice lens arising from this melting event? How deep did water percolate within this summer season? I would expect at least a paragraph dealing with such uncertainties, apart from the given uncertainty of 0.5a for layer dating, which represents a strange value dealing with IRHs generated from end-of-melt-season surfaces. • The layer picking remains a bit unclear. What happened for the 2011 IRH after Core 14? The indicated layer is almost horizontally flat, which certainly does not correspond to the layers underneath or above. Zooming in, I cannot follow the 2011 tracked reflection horizon. I would certainly pick the IRH from 2014? or 2010? layer instead, which are much more prominent. Can you comment on this? • Values in Section 2.2 are not correct. Here, you mixed up digits a bit. A RELATIVE DIELECTRIC (please consistently use this phrase) permittivity of 1.26 would correspond to a bulk density of \rho_s=145kg/m3, which is certainly not

the case for firn. Please correct accordingly and also correct the derived depth ranges. • There are several parts, where I would like to see quantifications (e.g., L24, L132, L169ff, L475ff) • Thermistors in bore holes need to settle before they can provide reliable numbers. I can see that this is impossible for the field approach you chose but can you provide comparisons of thermistor with MODIS annual temps? You should at least mention difficulties of an open bore hole for temp data. • Please revise the manuscript carefully for punctuation marks. I found numerous missing commas.

---

## Referee Comment (RC2) · Anonymous Referee #2 · 5 Aug 2019

GENERAL OVERVIEW: Lewis et al. work titled "Recent Precipitation Decrease Across the Western Greenland Ice Sheet Percolation Zone" reconstructs annual accumulation rates by using a well-known method of combining snow/firn density profiles from ice cores with the depth at which radar isochrones are found; in the dry-snow zone, radar isochrones are related to the depth-hoar formed at the end of summer, effectively marking annual accumulation layers. Here, they use the methodology in the percolation zone, and compare results with those of regional climate models to conclude that precipitation rates in the percolation zone of western Greenland show a decreasing trend. The data presented is of interest, and the radar data obtained over the percolation zone is certainly of importance. The paper is well written and clear, and I have few corrections regarding that. The methodology is well described, but I

do however have some comments regarding the validity of the it given the interpretation of results. The paper could also do a better job summarizing recent studies in the area; this needs to be addressed to avoid any impression that authors are cherry-picking results to reinforce their conclusions. Only Overly et al. findings are quoted using a similar method, but there are several studies showing that accumulation rates are increasing in this area (e.g. Koenig et al.). There is too much emphasis on the comparison with IceBridge radars, but clear differences between the VHF pulse radars and microwave phase-sensitive radars must be made because they operate differently. Although the uncertainties in the shallow firn core data are well explained, there is not sufficient details on the radar uncertainties, which are definitely large enough. This can even be seen at the sites where the shallow firn cores were taken (e.g. Figure 5). In my opinion, the emphasis should not be so much decreasing accumulation, which the authors hypothesize is caused in part by blocking of storms in the summer; the models only show a very slight decrease when looking at decadal trends, and the differences with the radar-estimated rates are larger than that, even over the core sites (Figure 5). I have specific comments as well that should be addressed before this paper is accepted. SPECIFIC COMMENTS: Section 2.2. How do you differentiate between annual accumulation layers (depth hoar formed in September/October) from percolation layers formed during the summer? As stated, unlike phase-sensitive radars, GSSI pulse radars can penetrate ice layers if they are thin enough, but without power analysis they look the same as depth hoar. Ln 157 A radar isochrone is by definition continuous IRHs, so this is redundant. What you really mean is that the isochrones observed have been related to annual accumulation layers. Ln 233 Why is the diameter needed? isn't the diameter of the cores approximately the same? If this is due to irregularities in the shape core, then it has to be explained that the core is assumed to have a cylinder-like shape with measured diameter. Ln 253-254. This phrase is not clear; please explain better. Ln 256-260. It is really hard to believe this statement without more in-situ data. As a matter of fact, there are studies that show that 21st Century percolation facies not only consist of pipes and lenses, but widespread layers that do amount to a fraction of

the total accumulation (Perry et al., 2007; Helm et al., 2006; de la Pena et al., 2015; Machguth et al., 2016). At the very least, an assessment of the uncertainties related to this should be given. Section 2.4. Is this different as what is shown in Figure 2? Section 2.2 states a constant dielectric to estimate depth. Please clarify. Section 2.5. It is stated that sometimes a "layer appears to bifurcate...". How does the authors know that the layer being traced is an actual annual layer (e.g. a depth hoar) and not a percolation feature? Ln 313-318. If the range resolution of the radar as stated in Section 2.2 is 0.35 m, then how it is possible that two radar samples are 0.12 m? This is inconsistent. My guess is that the uncertainty in accumulation estimates just from this would be at least the resolution times density, which is much higher than what is stated here. Ln 325-326. But it was stated in Section 2.3. that variable percolation facies do not affect estimates. I know is further discussed in Section 3.5, but my opinion is that more emphasis should be made in the variable structure of firn over the percolation zone. Ln 673-674. Please provide references. Ln 677-678. I do not believe this statement is correct. Uncertainties in radar-derived rates are in my opinion much larger. Figure 1. Please include elevation contour lines, it would be helpful for the reader even if most of the traverse is along an elevation of 2100 masl. Figure 5. Please add error bars to the GPR-estimated accumulation. Figure 5 and 12. Please use a larger font size.

---

## Author Comment (AC1) · 2 Sep 2019

The manuscript "Recent Precipitation Decrease Across the Western Greenland Ice Sheet Percolation Zone" by Lewis et al. presents large scale GPR transects and accumulation derivations thereof for more than 4400km of the Western GrIS. Such data are combined with firn cores to enable layer dating and accumulation calculations from density measurements. Vertical in-situ data allow accumulation derivations for the last 2 to 6 decades enabling trend assessments. In-situ trends are compared with RCM outputs to analyze for changes in accumulation and precipitation in relation with global temperature changes. The authors describe significant decreases in accumulation rates within the last 2 decades, which they attribute to shifting storm tracks reducing precipitation mainly for the summer months and increasing surface melt. I consider the presented work as novel and certainly significant for the scientific community especially because of the extensive data collection presented in this work. However, some redundancies, imprecise descriptions and the confusing structure of the manuscript prevent publication in the current state. I recommend to focus more on conciseness and maybe reconsider the total volume of the presented data. How about splitting into 2 manuscripts: one presenting the in-situ data including validation/ comparison with RCM results and the subsequent dealing with implications and atmospheric circulation simulations. Right now, the reader gets a bit lost in all the error/ uncertainty analyses combined with validation proofs for numerous statistics.

**Thank you for your thorough and helpful review. We appreciate that the manuscript covers a great deal of ground between the extensive data collection and climate-based analysis. However, we have decided not to split the manuscript in order to keep the data collection and analysis together. We feel that the background and data collection are necessary to motivate the reader to think about recent GrIS SMB changes. We use the field measurements to validate RCMs, which we then use to examine widespread SMB changes across the whole GrIS. We do not think two manuscripts would be able to portray these important results as accurately as one longer manuscript.**

**We have shortened the manuscript and reduced the length of several sections, particularly the introduction, to reduce the total volume of information. We believe that the manuscript is more concise and will nicely fill a gap in the literature of recent GrIS SMB measurements.**

Major points of criticism are: The structure of the manuscript is very confusing. The methods section comprises large fractions of discussion and data interpretation. Please revise the structure and attempt to shorten the manuscript whenever possible.

**We agree that some of the text originally in the methods is too verbose and is not appropriate for this section. Specifically, we removed material about the average relative permittivity, clarified how meltwater percolation effects isotope signals, added a sentence about comparing thermistor measurements with MODIS satellite derived temperatures, and removed a sentence within the leave-one-out cross validation paragraph. We feel that the radargram and density plots, while technically results from this study, belong in the methods section because they help the reader better understand the accumulation calculations and TWT-depth conversions.**

The introduction comprises almost 3 pages. It is clear to me that you want to introduce all relevant literature and topics, which are presented. However, if splitting into 2 manuscripts (see above), you could certainly focus more on less different topics. Parts, which could be shortened are L54ff and L89ff.

**We have shortened the introduction from three pages to two pages and removed unnecessary background material. We have shortened much of the material discussed in L54-L89 because Greenland summertime melting has previously been thoroughly discussed in the literature and does not need to be explained here in great detail. We appreciate this feedback and feel it has made the manuscript more concise.**

At least to me, it remains unclear how specific values are determined. For instance, epoch and annual accumulation values are hard to distinguish. It would be better to clearly distinguish in between these two.

**Annual accumulation is determined from the firn core chemistry data and is only shown in the background of Figure 5. We do not use discuss individual annual accumulation rates in this manuscript.**

**Epoch accumulation (average accumulation over multiple years) is calculated from adjacent IRHs (equation 3) in the geophysical data across the entire GreenTrACS traverse. We use these values to determine changes in accumulation in sections 3.4 and 3.5.**

**We changed L330 to "We assume uncertainty in dating the firn cores from annual variations in chemistry…" to clarify this point.**

Did you actually pick each individual layer in the radar data or just for specific locations where layer resolution is clear or just the 5 year layers as indicated in Fig. 5? This remains unclear, same for the accumulation calculations.

**As can be seen in Figure 3, we trace these individual layers across the dataset wherever the penetration depth (and equipment malfunctions) allows us to do so. Accumulation is calculated everywhere along the GreenTrACS traverse.**

**We have added to L269-271: "Each horizon is traced throughout the traverse, except in areas where the attenuated signal makes it too difficult to interpret."**

You state that 1m fractions as well as 3cm parts of the cores are analyzed (L233ff) in the field and lab. Were those core fragments further cut for more highly resolved density measurements?

**Core fragments were measured and weighed in the field as well as in the Dartmouth College Ice Core Laboratory freezer to calculate depth-density profiles. We repeat these measurements in case cores are lost or melted in transit, to double check for measurement errors, and to reacquire measurements in a controlled laboratory setting. We measured pieces along natural core breaks during the drilling process and did not further cut these pieces for higher resolved density measurements. For more information see Graeter et al. (2018).**

In addition, average melt rates in Fig. 11 and discussed in Section 3.5 are not adequately explained. I don't see how such values are generated (derived from RCMs, calculated in accordance to observed ice lenses as in L581?).

**Melt rates were measured from collected firn cores and published in Graeter et al. (2018). We measured ice layer thickness for each core using a light table in the Dartmouth College Ice Core Laboratory freezer. We then total the ice layer thickness per year using the chemistry derived depth-age scales.**

**We have added text to L213-214: "We measured melt layer thickness in the laboratory following Graeter et al. (2018)."**

RMS values describing deviations from RCMs lack an explanation for the uncertainty range. **We have added the following text to L466-468 "Averaged over all 4436 km of the traverse, the RMS difference (± 1σ) between each model and GreenTrACS accumulation over corresponding data periods…"**

In summary, I must admit, I got lost with all the uncertainty values being presented. What are sigma_epoch errors, how are these values related to sigma_accumulation-rate? I recommend to work carefully on the respective sections and maybe include a sketch of the applied workflow to derive accumulation data from radar IRHs.

**$\sigma_{epoch}$ is the uncertainty in accumulation rate for any single epoch. This combines all the individual uncertainties discussed in section 2.6 into one general uncertainty that we can use to compare our accumulation rate for a specific epoch with RCM accumulation rates. $\sigma_{n-epochs}$ is the uncertainty in accumulation rate for multiple epochs. We use this uncertainty when comparing our accumulation rate over multiple epochs with RCM accumulation rates.**

**We have clarified equations 5 and 6 to simplify these complicated concepts. L341-350 now reads "We find the total accumulation rate uncertainty for each epoch to be 0.0709 m w.e. a$^{-1}$ from equation 5.**

$$\sigma_{epoch} = \sqrt{\dot{b}^2 \left( \left( \frac{\delta h}{\Delta h} \right)^2 + \left( \frac{\delta t}{\Delta t} \right)^2 + \left( \frac{\delta \rho}{\rho} \right)^2 \right)} \qquad (1)$$

**… To calculate uncertainty for accumulation averaged over multiple epochs ($\sigma_{n-epochs}$) we divide our uncertainty $\sigma_{epoch}$ by the square root of the number of traced layers (n) at that location.**

$$\sigma_{n-epochs} = \frac{\sigma_{epoch}}{\sqrt{n}} \qquad (2)."$$

Just for clarification: The accumulation rate uncertainty is 71kg/m2/a, I interpret this value as the max accuracy you can achieve from GPR transects The RMS deviation to IceBridge accumulation rates is 39kg/m2/a, which is within the error margins. For annual accumulation rates in Fig. 5, I would expect to have error margins as stated above being included. How reliable is a 5-year standard deviation in accumulation rates?

**Thank you for the clarification question. The GPR accumulation rate uncertainty for any single epoch is 0.0709 m w.e. a$^{-1}$ and the average RMS difference from IceBridge accumulation rate is 0.0387 m w.e. a$^{-1}$, so they are statistically indistinguishable from each another.**

**The error bars in Figure 5 represent those uncertainties.**

**The five-year standard deviation in firn core accumulation rates accurately captures the variability of year-to-year fluctuations in accumulation throughout this region.**

The RMS deviation to RCMs is 48-82kg/m2/a and again within the error margins of the radar. Annual trends in precip are at 7kg/m2/a2. Consequently, you would need at least a 10 year period to reach the error margins for deriving trends, right?

**You are correct in that the average RMS difference from RCM accumulation is 0.0475 to 0.0822 w.e. a$^{-1}$, although these differences are much larger in certain regions of the traverse (see Figure 9).**

**Our GPR accumulation trends are 0.009 ± 0.005 m w.e a⁻² from 1996 to 2017, while RCM accumulation trends are 0.0016 to 0.003 m w.e a⁻² larger than that. While these trends are an order of magnitude smaller than the RMS difference between GPR and RCM accumulation, we have shown both the validity of our measurements and their agreement with RCM trends. Therefore, we are confident that both our measured trends and RCM trends exist.**

How is the vertical resolution limit of the 400MHz antenna calculated? For firn of roh_s=550 kg/m3 you would receive a v_mean of 0.2m/ns resulting in a wavelength of 0.5 m. Resolution limits are sometimes defined as half of the wavelength or 1/4*lambda. How do you come up with 0.35m?

**The interface separation resolution is defined by the bandwidth, which controls the pulse duration, and not the center frequency (see Appendix C of Marshall and Koh., 2008, which is applicable to both FMCW and impulse radars). GPR systems usually have a bandwidth on the order of the center frequency. For a velocity of 0.2 m ns⁻¹, we can use the equation for range resolution, v/(2*bandwidth) = 25 cm. We could not distinguish separate features within less than 0.35 m in our radargrams, so we conservatively choose a resolution limit of 0.35 m.**

You discuss several times errors introduced by percolating melt water. Heilig et al. (2018) measured the seasonal mass flux from snow into underlying firn at Raven to be at >50kg/m2 (in your preferred units >0.05m w.e.) for summer 2016. Can you clearly date back ice lenses or is the mentioned ice lens from 2003/04 a result of several melt seasons? What about summer 2012? Shouldn't there be a thicker ice lens arising from this melting event? How deep did water percolate within this summer season? I would expect at least a paragraph dealing with such uncertainties, apart from the given uncertainty of 0.5a for layer dating, which represents a strange value dealing with IRHs generated from end-of-melt-season surfaces.

**We cannot be confident dating the ice lenses to a particular year, as meltwater typically percolates to depths greater than 1 m (Benson, 1962; Cox et al., 2015; Harper et al., 2012). The ice layer located within any given year may have been generated from that year or a following year. However, we can confidently date the surrounding snow, as the oxygen isotope and major ion signals remains relatively unperturbed (see Neff et al., 2012 – Journal of Glaciology; Avak et al., 2018 – Journal of Glaciology).**

**We have updated the text to "While meltwater percolation smooths the signal of some of these tracers, we can still confidently determine the depth-age curve using nearly-unperturbed oscillations in δ¹⁸O and dust."**

**The ice lens from the 2012 event is likely thicker throughout the traverse than ice lenses from other summers. We can still confidently calculate SMB over 5 year periods from this method by analyzing the amount of mass between adjacent IRHs.**

**The 0.5 year uncertainty arises from dating the firn core using isotope and major ion chemistry, not from counting IRHs annual layers like Medley et al. (2013) or Koenig et al. (2016).**

The layer picking remains a bit unclear. What happened for the 2011 IRH after Core 14? The indicated layer is almost horizontally flat, which certainly does not correspond to the layers underneath or above. Zooming in, I cannot follow the 2011 tracked reflection horizon. I would

certainly pick the IRH from 2014? or 2010? layer instead, which are much more prominent. Can you comment on this?

**The resolution of this image is too low to clearly see the undulating IRHs along the 2011 layer. We have double checked the layer picks in Figure 2 and observed a small error in the 2011 layer. We have fixed that IRH and recalculated accumulation across that region, noting that none of the accumulation measurements change by more than 0.01 m w.e. a⁻¹.** After reexamining the rest of our layer picks, we are confident that they are correct. Note that we will publish both our GPR data and layer picks with this manuscript so that others can verify our interpretation of the data.

**This image serves as a subset of the traced IRHs from the entire 2017 traverse to highlight the high spatial resolution of our dataset. We purposefully traced these IRHs throughout the dataset rather than tracing specific prominent horizons for short distances.**

Values in Section 2.2 are not correct. Here, you mixed up digits a bit. A RELATIVE DIELECTRIC (please consistently use this phrase) permittivity of 1.26 would correspond to a bulk density of rho_s=145kg/m3, which is certainly not the case for firn. Please correct accordingly and also correct the derived depth ranges.

**Thank you for pointing out this error. We have corrected the usage to "relative dielectric permittivity" throughout the manuscript.**

**We have removed this sentence entirely as it is confusing to the reader. The derived depth ranges were not calculated using a constant relative dielectric permittivity, and are not affected by this error.**

There are several parts, where I would like to see quantifications (e.g., L24, L132, L169, L475).

**We have added quantifications to these locations to indicate the recent decrease in accumulation. The text now reads "…show decreasing accumulation and precipitation of 2.4 ± 1.5 % a⁻¹" and is easier to understand.**

Thermistors in boreholes need to settle before they can provide reliable numbers. I can see that this is impossible for the field approach you chose but can you provide comparisons of thermistor with MODIS annual temps? You should at least mention difficulties of an open borehole for temp data.

**Correct that borehole thermometry is usually conducted over periods longer than 24-48 hours. However, the thermistor at 20 m depth (thick black line on figure below) is able to asymptotically equilibrate within 24 hours to within ±0.1 °C and provides a temperature that we are confident can be used to drive a Herron-Langway density profile. Please see an example of the data from Core 14 below.**

**We added the following text to L225: "These measurements agree with MODIS satellite derived mean annual temperature (Hall et al., 2012) to within ±1 °C for each firn core location."**

[Figure]

Please revise the manuscript carefully for punctuation marks. I found numerous missing commas.

**The manuscript has been revised for missing commas.**

---

## Author Comment (AC2) · 2 Sep 2019

GENERAL OVERVIEW: Lewis et al. work titled "Recent Precipitation Decrease Across the Western Greenland Ice Sheet Percolation Zone" reconstructs annual accumulation rates by using a well-known method of combining snow/firn density profiles from ice cores with the depth at which radar isochrones are found; in the dry-snow zone, radar isochrones are related to the depth-hoar formed at the end of summer, effectively marking annual accumulation layers. Here, they use the methodology in the percolation zone, and compare results with those of regional climate models to conclude that precipitation rates in the percolation zone of western Greenland show a decreasing trend. The data presented is of interest, and the radar data obtained over the percolation zone is certainly of importance. The paper is well written and clear, and I have few corrections regarding that. The methodology is well described, but I do however have some comments regarding the validity of the it given the interpretation of results. The paper could also do a better job summarizing recent studies in the area; this needs to be addressed to avoid any impression that authors are cherrypicking results to reinforce their conclusions. Only Overly et al. findings are quoted using a similar method, but there are several studies showing that accumulation rates are increasing in this area (e.g. Koenig et al.).

**Thank you for your review and comments, we believe they have made the manuscript stronger and more succinct. Our introduction covers all recent *in situ* radar studies in this region and we are the first to collect data throughout many regions in the traverse. We highlight several studies that use similar methods (e.g. Hawley et al., 2014) and studies using other methods that found different results (e.g. Wong et al., 2015; Overly et al., 2016).**

**Our results are statistically indistinguishable from those of Koenig et al. (2016; not shown) over 2009-2012. Our accumulation trends from 1996-2016 cover a longer duration than the data from that study and their accumulation trends are almost all statistically insignificant. Koenig et al. (2016) discusses increased accumulation near Camp Century only within the MAR RCM, which "differ in magnitude from the radar-derived measurements in 2010 or 2011."**

**The following text has been added to section 3.2 "Similarly, our 2011-2016 accumulation is statistically indistinguishable from average 2009 – 2012 IceBridge snow radar measurements analyzed by Koenig et al. (2016), with an RMS difference of 0.0489 ± 0.0961 m w.e. a$^{-1}$ along a total of 69.7 km of overlap (not shown). Koenig et al. (2016) use a different radar system on an airborne platform and are able to calculate annual accumulation at elevations below 2000 m a.s.l., however the GreenTrACS accumulation record covers a much longer temporal duration than the data from that study."**

There is too much emphasis on the comparison with IceBridge radars, but clear differences between the VHF pulse radars and microwave phase-sensitive radars must be made because they operate differently.

**The following text has been added to the introduction "Note that our *in situ* GPR operates using a UHF pulsed radar, while other systems such as frequency modulated continuous wave (FMCW) radars use phase-sensitive radar architecture that include both amplitude and phase information." While the pulse and phase-sensitive radars operate differently, the radargrams generated by pulse radars within the VHF-UHF spectrum allow us to trace isochronous IRHs and calculate accumulation, in a similar way to the airborne FMCW approach. In dry snow/firn, the relative dielectric permittivity is not sensitive to frequency in the range between UHF and microwave, and therefore the radar velocity is not influenced by the different frequencies of these systems.**

Although the uncertainties in the shallow firn core data are well explained, there is not sufficient details on the radar uncertainties, which are definitely large enough. This can even be seen at the sites where the shallow firn cores were taken (e.g. Figure 5).

**A detailed explanation of the radar uncertainty can be found in Section 2.6.**

**Figure 5 shows that radar and firn core accumulation measurements are statistically indistinguishable at four example core sites, which is also the case at all sixteen core locations. We believe that these uncertainties are small enough to allow for analysis of accumulation trends in our dataset.**

In my opinion, the emphasis should not be so much decreasing accumulation, which the authors hypothesize is caused in part by blocking of storms in the summer; the models only show a very slight decrease when looking at decadal trends, and the differences with the radar-estimated rates are larger than that, even over the core sites (Figure 5).

**Although the decrease in accumulation is small, we show throughout the manuscript that it is not negligible. Furthermore, the general narrative in the literature is that accumulation is increasing, and will continue to increase, with higher human-forced temperatures due to higher saturation vapor pressures. We show that this is not the case over the past two decades in our study region, and we point to the importance of summer blocking as a driver of the accumulation decline, which has not been discussed extensively in the literature. We emphasize the accumulation decline because none of the CMIP5 GCMs can accurately capture recent Greenland blocking activity (Hanna et al., 2018), and our results highlight that mass loss is currently occurring from both sides of the SMB equation (declining mass input, and accelerating mass output from melting and runoff). We therefore respectfully disagree that this should not be emphasized in the paper; we believe that it is the most important contribution that this paper makes to our understanding of Greenland SMB.**

Section 2.2. How do you differentiate between annual accumulation layers (depth hoar formed in September/October) from percolation layers formed during the summer? As stated, unlike phase-sensitive radars, GSSI pulse radars can penetrate ice layers if they are thin enough, but without power analysis they look the same as depth hoar.

**We do not differentiate between annual accumulation and percolation layers. Rather, we calculate accumulation between adjacent IRHs using the age and mass between these isochrones, determined from the depth-age scales and densities interpolated from the firn cores. The SMB is indifferent to where the mass originated, all we're trying to do is calculate that mass balance.**

Ln 157 A radar isochrone is by definition continuous IRHs, so this is redundant. What you really mean is that the isochrones observed have been related to annual accumulation layers.

**We have updated the text to reflect this distinction. The text now reads "The 400 MHz short-pulse radar has a range resolution (ability to resolve distinct features) of $0.35 \pm 0.1$ m in firn, which is fine enough to resolve Internal Reflecting Horizons (IRHs) that have been related to annual accumulation layers (Medley et al., 2013; Rodriguez-Morales et al., 2014; Spikes et al., 2004; Hawley et al., 2014)."**

Ln 233 Why is the diameter needed? isn't the diameter of the cores approximately the same? If this is due to irregularities in the shape core, then it has to be explained that the core is assumed to have a cylinder-like shape with measured diameter.

**The diameter of the core fluctuates slightly (<1 mm), so to accurately calculate the volume and density of each core section we measure the diameter of the core at the beginning, middle, and end of that section using calipers. Since the radius is squared in the cylinder's volume calculation, it is imperative to know the radius as accurately as possible for density calculations. For more information see Graeter et al. (2018).**

Ln 253-254. This phrase is not clear; please explain better.

**The text has been modified to "Final calculated accumulation rates are insensitive to the input accumulation parameter we use to calculate our Herron-Langway models (Lewis et al., 2017)."**

Ln 256-260. It is really hard to believe this statement without more in-situ data. As a matter of fact, there are studies that show that 21st Century percolation facies not only consist of pipes and lenses, but widespread layers that do amount to a fraction of the total accumulation (Perry et al., 2007; Helm et al.,

2006; de la Pena et al., 2015; Machguth et al., 2016). At the very least, an assessment of the uncertainties related to this should be given.

**These studies are all from lower elevations on the ice sheet, where certainly the reviewer is correct that ice lenses can be widespread and account for a significant fraction of the year's total accumulation. At the higher elevation of our firn cores, however, we did not observe widespread ice lenses across the snow pits used to extract cores, snow pits used for stratigraphic analysis, or snow pits used for camp. Cores 1 – 7 had an average of 1 – 5 cm total ice layer thickness per year, while cores 9 – 16 had less than 2 cm of melt per year, most of which occurred during the past decade.**

Section 2.4. Is this different as what is shown in Figure 2? Section 2.2 states a constant dielectric to estimate depth. Please clarify.

**We have removed the sentence in Section 2.2 that made it appear we were using a constant relative dielectric permittivity to estimate depth. In actuality, we calculate permittivity from the density (equation 2) in order to calculate the velocity (equation 1) so that we can determine depth from the TWT.**

Section 2.5. It is stated that sometimes a "layer appears to bifurcate…" How does the authors know that the layer being traced is an actual annual layer (e.g. a depth hoar) and not a percolation feature?

**We do not distinguish between annual layers and percolation features, rather, we trace IRHs from one firn core to another in order to calculate SMB between the two cores. It doesn't matter what contrast in relative dielectric permittivity is causing the IRH, all that matters is that these horizons are isochronous and we know the date of each layer within ±0.5 years. If the accumulation rate changes substantially and layers bifurcate multiple times, it would be possible that the traced IRH represents a different part of the year from the original traced layer. Since our epochs represent five years, at most, this could change the length of the epoch by ~10%, but we do not have any IRHs between adjacent firn cores that exhibit this behavior.**

Ln 313-318. If the range resolution of the radar as stated in Section 2.2 is 0.35 m, then how it is possible that two radar samples are 0.12 m? This is inconsistent. My guess is that the uncertainty in accumulation estimates just from this would be at least the resolution times density, which is much higher than what is stated here.

**The range resolution (ability to distinguish distinct features) is 0.35 m, and is controlled by the radar bandwidth, but the radar sample spacing, which is controlled by the sample frequency of the analog to digital converter, is 0.12 m. We cannot definitively distinguish which range bin the IRH lies within, hence our uncertainty of 0.35 m. The resulting uncertainty in accumulation is 0.0709 m w.e. a$^{-1}$, accounting for uncertainties in radar precision, tracing IRHs, errors in dating the firn cores, and errors in our density estimates.**

Ln 325-326. But it was stated in Section 2.3. that variable percolation facies do not affect estimates. I know is further discussed in Section 3.5, but my opinion is that more emphasis should be made in the variable structure of firn over the percolation zone.

**In this paragraph we are saying that the difference between calculating accumulation using measured density profiles and calculating accumulation using estimated/interpolated density profiles has larger errors for the southern cores because meltwater percolation and ice lenses complicate the density profile. We have added the following text to L316-317 "Throughout this study, we use our measured density profiles to calculate accumulation at core locations, rather than rely on Herron-Langway density models that would result in larger uncertainties."**
**Numerous studies have documented the heterogeneity of firn throughout the percolation zone and the complications of calculating SMB due to ice pipes and lenses. Here we attempt to accurately calculate accumulation using firn cores and *in situ* GPR throughout this complicated region. The text has been updated to reflect these complications.**

Ln 673-674. Please provide references.

**We have added references for these climate models. This sentence now reads "Overall, the Polar MM5 (Burgess et al., 2010), MAR (Fettweis et al., 2016), Box13 (Box et al., 2013), and RACMO2 (Noël et al., 2018) Regional Climate Models accurately capture large spatial patterns in accumulation over the GrIS, but show statistically significant differences from GPR accumulation on a regional basis."**

Ln 677-678. I do not believe this statement is correct. Uncertainties in radar-derived rates are in my opinion much larger.

**Please see section 2.6, and specifically equation 5, for formal error propagation and uncertainty calculations. We believe that we have done everything to accurately constrain the accuracy of this radar system and have been conservative in our uncertainty analysis. For comparison, Hawley et al. (2014) calculate an accumulation uncertainty of ~0.015 m w.e. a$^{-1}$ using a similar geophysical system, Overly et al. (2016) calculate an accumulation uncertainty of 0.01 m w.e. a$^{-1}$ using the ASIRAS airborne radar, and Medley et al. (2013) calculate an accumulation uncertainty of 0.055 m w.e. a$^{-1}$ using the IceBridge snow radar. Our total accumulation rate uncertainty for each epoch of 0.07 m w.e. a$^{-1}$ is the same order of magnitude, but larger, than those reported uncertainties.**

Figure 1. Please include elevation contour lines, it would be helpful for the reader even if most of the traverse is along an elevation of 2100 masl.

**We have added the 2000 m and 3000 m contour lines to Figure 1. We believe these give an idea of the elevation of our traverse without crowing the figure too much.**

Figure 5. Please add error bars to the GPR-estimated accumulation.

**Error bars in the GPR accumulation are indicated in red. We do not show error bars for the annual core accumulation to simplify the figure, however the error bars for the 5 year averaged core accumulation is indicated in black (GreenTrACS cores) and blue (PARCA cores).**

Figure 5 and 12. Please use a larger font size.

**We have increased the font size for both figures. These figures are now easier to understand.**

---

## Author Response (AR2)

I reevaluated the manuscript submitted by Lewis et al. entitled "Recent Precipitation Decrease Across the Western Greenland Ice Sheet Percolation Zone". The structure of the paper improved and through removal of several parts the reader gets less distracted by numerous presented uncertainty values and parameters. I can support publication after some minor - mostly technical corrections.

However, scientifically, I still have concerns about Fig. 11 and conclusions derived from it. According to my understanding, a causal relationship in between trends in SMB and average melt rates is not necessarily the case. A strong accumulation year followed by a strong melt season could still result in average SMB values. The trend in SMB would be unaffected but the average melt rate increased. Since a causality between average melt and trends in SMB is not present, a linear regression for the given significance level does not allow interpretations such as the ones being presented. In addition, only 1/3 of the points are within the confidence bounds, while e.g. strong melt and a strong negative trend in SMB can occur (C1) same as low melt and an even stronger negative trend in SMB (C8). Same occurs for average trends. Melt rates for roughly a trend of $-6 \times 10^{-3}$ m w.e. $a^{-2}$ are within a range of 0 - 0.11 m w.e. per year in melt. I am quite skeptical concerning the statements in L557-564 as well. I cannot see a confirmation of the hypothesis that percolation and refreezing is enhancing the negative accumulation trends. I know, it would be very intuitive but the cores you present do not show this. For a trend "above" -0.006 m w.e. $a^{-2}$, 4 cores result in an average melt rate below 0.04 m w.e. per year and 4 cores are situated at or above 0.08 m w.e. melt per year. I recommend to remove Figure 11 and the corresponding text in L557-564. Especially, since the following lines are contradictory to the statement that melt influences trends within the 20a period.

**We would like to thank the reviewer for their time reevaluating this manuscript and for continuing to improve the scientific quality of this paper.**

**We agree that Figure 11 distracts from the focus of the manuscript. We have removed Figure 11 and L557-564.**

Some technical details:
Please present significant digits and be consistent with it. Several occurrences of statements with different levels of accuracy are placed even within given ranges (e.g. L502). I don't see the necessity to present sub mm accuracies especially while accuracy levels do not allow for such precision.

**Thank you for noticing this oversight.**

**We have modified the manuscript to have a consistent number significant figures.**

You should differentiate in between accumulation and accumulation rates. Within the latter part of the manuscript (Section 3), you almost exclusively use the term accumulation for given accumulation rates. Be consistent!

**We have updated "accumulation" to "accumulation rates," where appropriate, throughout the manuscript**

Please carefully correct typos, edits and missing links:
L270-275: 4 sentences in a row start exactly equally

**We have modified the beginning of the sentences to avoid repetition**

L101 link missing

**Link has been updated**

I recommend to include more often percentage values especially for trends and errors. This facilitates the assessments of errors and trends especially for accumulation rates.

**We have modified L535-536 to include percentage values: "On average, the RCMs have a more**

**negative precipitation trend than the GreenTrACS record by 0.003 ± 0.005 m w.e. a$^{-2}$ (0.3 ± 0.77%) for MAR and 0.002 ± 0.005 m w.e. a$^{-2}$ (0.45 ± 1.22%) for RACMO2."**
**We have modified L413-415 to include percentage values: "Average (1966 – 2016) GPR accumulation rates are statistically indistinguishable with average (1962 – 2014) IceBridge Accumulation Radar measurements analyzed by Lewis et al. (2017), with an RMS difference of 0.039 ± 0.033 m w.e. a$^{-1}$ (6.0 ± 9.6%) along a total of 562.5 km of overlap"**
**We have modified L425-427 to include percentage values: "Similarly, our 2011-2016 accumulation rates are statistically indistinguishable from average 2009 – 2012 IceBridge snow radar measurements analyzed by Koenig et al. (2016), with an RMS difference of 0.049 ± 0.096 m w.e. a$^{-1}$ (14.0 ± 27.7%) along a total of 69.7 km of overlap (not shown). "**

L312 ...leave-one-out...
**This typo has been fixed to "…leave-one-out validation…"**

L340 missing unit for Delta-h
**We have added units to read "$\Delta h$ = 3.56 m"**

L383 I disagree with higher accumulation rates in the SW. I would rather identify higher rates in the central parts of the transects.
**We have modified the text to read "…with higher accumulation rates along the main traverse and lower accumulation rates at higher elevations…"**

L399 ...firn are within error of those... should be ... are within uncertainty ranges of those...?
**We have modified the text to read "Accumulation rates derived from GreenTrACS firn cores are within uncertainty ranges of those…"**

I don't see the benefit of the plotted annual accumulation rates in Fig. 5. As far as I remember, those rates are not quoted in the manuscript either.
**We use these annual accumulation rates to show the variability in year-to-year accumulation and the benefit of averaging accumulation rates over 5 year periods.**
**We have modified the text on L 398-399 to read "Annual and epoch-averaged accumulation rates derived from GreenTrACS firn cores are within uncertainty ranges…"**
**Additionally, we have added the following text "Averaging accumulation rates over five year epochs reduces noise in year-to-year accumulation variability."**

L341 give values for average firn density
**We have added the text "average firn density $\rho$ = 0.55 g cm$^3$,"**

Additional note: A recent TCD manuscript presents values for snow densities at Dye-2 for spring 2015 and the following years. In case you want to simplify your core assessment for Dye-2 consider using those.
Heilig, A., Eisen, O., Schneebeli, M., MacFerrin, M., Stevens, C. M., Vandecrux, B., and Steffen, K.: Spatial and temporal variability of snow accumulation for the South-Western Greenland Ice Sheet, The Cryosphere Discuss., https://doi.org/10.5194/tc-2019-184, in review, 2019.
**Thank you for pointing out this new and interesting publication.**
**At Dye-2, we use our firn core densities from Vandecrus et al., (2018) for the top 19.3 m and firn core densities from Bales et al. (2009) for depths between 19.3 and 119.6 m. Near-surface density at Dye-2 has not changed enough between 2015 and 2019 to alter our accumulation results.**

Finally, we have added a brief acknowledgement section that reads as follows:

"This project was supported by the US National Science Foundation (NSF) under grants DGE-1313911 and ARC-1417640. We would like to thank Mary Albert for providing field validation measurements, as well as Jason Box, Xavier Fettweis, and Brice Noel for providing the most recent Box13, MAR, and RACMO regional climate model outputs. Our successful data collection would not have been possible without the support of Ch2M Hill Polar Field Services, Kangerlussuaq International Science Support, and the Air National Guard 109th Airlift Wing. We thank the U.S. Ice Drilling Program for support activities through NSF Cooperative Agreement 1836328. Special thanks to Sean Birkel and the Danish Meteorological Institute for location-specific weather forecasts in Greenland. The authors would like to thank two anonymous reviewers for greatly improving the manuscript."

---

## Author Response (AR3)

Dear Dr. Lewis,
Thank you for submitting the revised version of your MS, which is now accepted for publication in TC, pending some technical corrections listed below.
Best wishes,
Michiel van den Broeke

Technical corrections
Technical corrections Lewis et al.

**Thank you for catching these typos and correcting the small errors you found. We have taken your comments into consideration and made the following changes to our manuscript.**
**Thank you for all the help,**
**Gabriel Lewis**

l. 19: remove 'critical'
**We removed "critical" from this sentence.**

l. 24: "show decreasing accumulation rates and precipitation of…" unclear, is it accumulation or precipitation that is decreasing, or both? Please reformulate.
**We removed "and precipitation" from this sentence. The text now reads "Trends from both radar and firn cores, as well as commonly used regional climate models, show decreasing accumulation rates of 2.4 ± 1.5 % a$^{-1}$ over the 1996 – 2016 period…"**

l. 27: high -> strong
**We changed "high" to "strong" in this sentence.**

l. 32: of -> to
**We changed "of" to "to" in this sentence.**

l. 34: GT -> Gt
**We changed "GT" to "Gt" in this sentence.**

l. 44: "rising Greenland summer temperature trends"; a rising trend means an acceleration. You probably mean a positive trend. Please reformulate.
**We changed "rising Greenland summer temperature trends" to "positive Greenland summer temperature trends."**

l. 53: warming temperatures -> increasing temperatures, also l. 678 and throughout, please.
**We changed "warming temperatures" to "increased temperatures" in L53 and L678.**

l. 68: off of -> off (?)
**We changed "off of" to "off" in this sentence.**

l. 224: "These measurements agree with MODIS satellite-derived mean annual temperature (Hall 225 et al., 2012) to within ±1 °C for each firn core location." Please add "surface" before temperature. This is unexpected for two reasons. First, MODIS only provides data for clear skies, and second one would expect that refreezing raises firn temperature compared to surface temperature. Please provide a brief statement to address this.

**We added "surface" before temperature.**

**We added the following sentence "The small amount of energy released from refreezing of summertime percolation water has diffused by the time of our measurements, allowing for direct comparison between _in situ_ firn temperature and MODIS clear sky measurements."**

l. 539: 2539.4 -> 2539 and likewise later I sentence.

**We changed "2539.4" to "2539" and "5159.1" to "5159" in this sentence.**

[revised manuscript text omitted]